# Logit-Attention Divergence: Mitigating Position Bias in Multi-Image Retrieval via Attention-Guided Calibration

**Mingtao Xian**[1] **Yifeng Yang**[‡ 2] **Qinying Gu**[3] **Xinbing Wang**[2] **Nanyang Ye**[2 3 4]

## Abstract

Multimodal Large Language Models (MLLMs) have shown strong performance in multi-image cross-modal retrieval, yet suffer from severe position bias, where predictions are dominated by input order rather than semantic relevance. Through empirical analysis, we identify a phenomenon termed Logit-Attention Divergence, in which output logits are heavily biased while internal attention maps remain well-aligned with relevant visual evidence. This observation reveals a fundamental limitation of existing logit-level calibration methods such as PriDe. Based on this insight, we propose a training-free, attention-guided debiasing framework that leverages intrinsic attention signals for instance-level correction at inference time, requiring only a minimal calibration set with negligible computational overhead. Experiments on MS-COCO-based benchmarks show that our method substantially improves permutation invariance and achieves state-of-the-art performance, enhancing accuracy by over 40% compared to baselines. Code is available at https://github.com/brightXian/LAD.

## 1. Introduction

Recent advances in Multimodal Large Language Models (MLLMs) have extended cross-modal reasoning to multi-image scenarios (Liu et al., 2024a; Song et al., 2024; Meng et al., 2025; Wang et al., 2025a). A critical task is in-context cross-modal retrieval, where models must identify and retrieve target images from sequences containing numerous distractors (Li et al., 2024; Wang et al., 2025a). This capability is crucial for real-world applications such as e-commerce

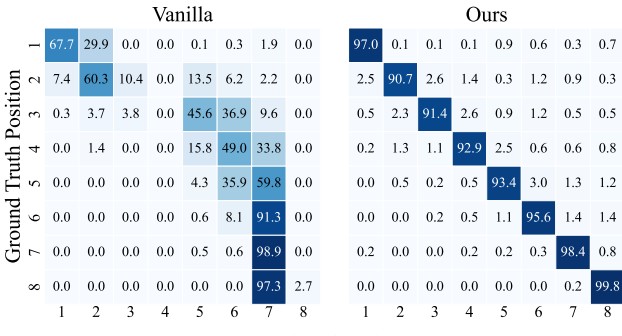

*Figure 1.* Confusion matrices for LLaVA-OneVision on the multi-image retrieval task ($N = 8$, Random setting). Cell values represent the selection rate (%), where darker blue indicates a higher selection rate. The Vanilla baseline (Left) shows vertical stripes due to severe position bias, while our method (Right) restores the diagonal pattern, demonstrating that predictions correctly align with the ground truth.

recommendation (Wang et al., 2025b; Tian et al., 2024), visual search in photo albums (You et al., 2024), and content moderation (Liang et al., 2025).

However, MLLMs exhibit severe position bias where predictions are dominated by input order rather than semantic relevance (Tan et al., 2024; Tian et al., 2025). As shown in Figure 1 (left), on an 8-candidate retrieval task, the model irrationally prefers certain positions (e.g., 48.96% selection rate at Position 7) while ignoring others (0.00% at Position 4). This bias leads to a drastic divergence in performance where accuracy is inflated to 98.87% when the ground truth coincides with the preferred Position 7, yet it collapses to 0.00% if the ground truth falls into the neglected Position 4. This position bias stems from the autoregressive architecture inherited from Large Language Models (Wu et al., 2025; Tian et al., 2025). While temporal positional encodings (Su et al., 2024; Vaswani et al., 2017) and causal attention masks (Radford et al., 2018; Touvron et al., 2023) preserve logical order during text generation, applying this architecture to non-temporal visual sequences introduces unintended asymmetries that render predictions highly sensitive to image input ordering due to structural position dependency.

Existing training-free interventions for position bias are predominantly developed for text-only LLMs (Peysakhovich & Lerer, 2023; Yu et al., 2024; Ratner et al., 2023; Wang

‡Project lead [1]Zhiyuan College, Shanghai Jiao Tong University, Shanghai, China [2]Shanghai Jiao Tong University, Shanghai, China [3]Shanghai Artificial Intelligence Laboratory, Shanghai, China [4]Shanghai Innovation Institute, Shanghai, China. Correspondence to: Nanyang Ye <ynylincoln@sjtu.edu.cn>.

*Proceedings of the 43rd International Conference on Machine Learning*, Seoul, South Korea. PMLR 306, 2026. Copyright 2026 by the author(s).

et al., 2025c). However, transferring these paradigms to MLLMs remains challenging as they overlook cross-modal interactions between vision and language. Our analysis in §3.1 reveals that position bias in MLLMs is not a static offset but a conditional bias that shifts dynamically with target location. This property renders existing methods ineffective: prompt-based approaches (Zheng et al., 2024; Wei et al., 2022) fail to adapt to such dynamic behavior, while static statistical calibration methods like PriDe (Zheng et al., 2024) rely on fixed priors that ignore the visually conditioned nature of position bias in MLLMs. In §3.2, we investigate the cross-modal interaction mechanisms of MLLMs and uncover a distinctive phenomenon:

> ***Due to position bias, MLLMs often "look at" the correct answer but "say" the wrong one.***

We term this discrepancy *Logit–Attention Divergence*. Concretely, while the final output logits are heavily corrupted by positional priors, the model's internal attention weights remain semantically aligned with the target visual region. Motivated by this insight, we propose the ***Attention-Guided Debiasing*** framework, a training-free framework that uses hierarchical attention to correct position bias at inference time. By bridging static positional priors with dynamic visual evidence, our method effectively corrects position bias **with only 5 calibration samples** and minimal additional computational cost. As shown in Figure 1 (right), it restores the diagonal confusion matrix, recovering permutation invariance.

**In summary, our contributions are as follows:**

- We identify a phenomenon termed *Logit-Attention Divergence*, where structural decoding priors override correct internal attention, causing models to "look right but speak wrong" in multi-image contexts.

- We propose Attention-Guided Debiasing, a training-free framework utilizing intrinsic visual signals to rectify position bias, requiring only a minimal calibration set (e.g., 5 samples) with negligible computational overhead.

- We construct an adversarial benchmark with visually similar hard negatives to test robustness under challenging conditions. Extensive experiments demonstrate state-of-the-art performance, effectively restoring permutation invariance across both random and adversarial settings, surpassing baselines by over 40% in accuracy.

## 2. Related Work

**Cross-modal Retrieval.** Early approaches to cross-modal retrieval rely on image–text embedding alignment, mapping both modalities into a shared space and performing retrieval via similarity matching (e.g., CLIP (Radford et al., 2021),

ALIGN (Jia et al., 2021), SigLIP (Zhai et al., 2023)). Although effective for coarse-grained matching, this approach struggles with fine-grained semantics as multimodal information is compressed into a single representation. With the advent of MLLMs, Li et al. (2024) introduced a novel generative retrieval paradigm that represents images as discrete token IDs, enabling MLLMs to memorize and directly generate the target image ID in response to a textual query. Building on this setting, Wang et al. (2025a) generalized the setting by instantiating retrieval as multi-image caption matching, where the MLLM is prompted to select the image that best matches a given caption by predicting a discrete index token.

**Position Bias in LLMs.** Similar to the position bias observed in LLMs (Zheng et al., 2024; Geng et al., 2024; Wang et al., 2024), recent work (Wang et al., 2025a) reveals that MLLMs also suffer from severe position dependency in multi-image caption matching. Specifically, MLLMs show a systematic preference for images placed at specific positions, particularly those appearing at the beginning or the end of the input sequence. Prior studies have attempted to mitigate this bias using explicit debiasing instructions (Zheng et al., 2024) appended to the system prompt, as well as Chain-of-Thought prompting (Wei et al., 2022) (e.g., "Let's think step by step"). However, experimental results indicate that neither approach effectively alleviates selection bias, suggesting that it is an inherent behavioral property of LLMs rather than an artifact that can be eliminated through simple prompt engineering.

**Inference-Time Calibration.** Inference-time calibration aims to adjust model behavior post hoc to correct model internal biases without modifying model parameters. Prior work generally falls into three categories. (1) Ensemble-based methods, such as self-consistency (Wang et al., 2022), aggregate predictions across multiple sampled outputs to improve robustness, implicitly assuming that biases cancel out through averaging. (2) Statistical calibration methods, like PriDe (Zheng et al., 2024), introduce a global static prior to reweight model logits. (3) Structure-aware approaches, such as SoFA (Tian et al., 2025), attempt to calibrate predictions by directly modifying the attention mechanism of LLMs without retraining, specifically by replacing causal attention with bidirectional attention. However, the modification introduces a mismatch between inference-time and training-time attention mechanisms, which may undermine the effectiveness and generality of MLLMs.

## 3. Motivation

### 3.1. Conditional Position Bias

Existing calibration methods, such as PriDe (Zheng et al., 2024), operate on the assumption that position bias is a

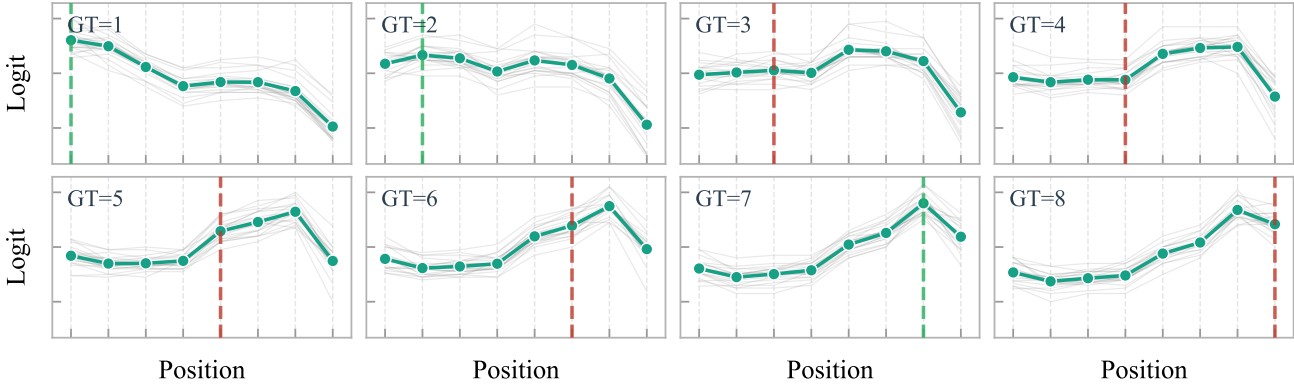

*Figure 2.* Logit distributions for candidate answer tokens conditioned on ground truth (GT) positions on LLaVA-OneVision. Each subplot shows a different GT position. Green curves represent mean logit profiles, gray lines show individual samples, and dashed lines mark the GT (green: correct; red: incorrect). Individual samples cluster tightly around the mean within each GT category.

static, content-independent offset. They estimate a global bias vector $P_{\text{prior}}$, which functions as a prior probability representing the model's intrinsic tendency to select specific positions independent of visual evidence. Consequently, these methods perform calibration by subtracting the logarithm of this static prior from the output logits. This additive correction in the log-domain is equivalent to dividing by the prior in the probability domain, factoring out the estimated frequency imbalances. However, our experiments in multi-image tasks reveal a critical flaw in this premise: position bias is not static but conditional. As shown in Figure 2, we statistically analyze the logit structures conditioned on varying ground truth (GT) positions. A distinct regularity is observed: within each GT category, the individual samples (gray lines) tightly cluster around the average trend (green curves). The final logit distributions at different GT positions exhibit stable and consistent structural profiles.

This observation exposes the limitation of static calibration methods like PriDe, which rely on subtracting a single global prior vector $P_{\text{prior}}$. As illustrated, the distributions for distinct targets (e.g., GT 6, 7, and 8) exhibit a phenomenon of homogenization, sharing nearly identical profiles. Since the global $P_{\text{prior}}$ represents a global average across all positions, subtracting this constant offset from these highly similar distributions fails to introduce discriminability. Consequently, the debiased logits remain statistically indistinguishable, proving that a global prior is insufficient to resolve such conditional ambiguity.

### 3.2. The Logit-Attention Divergence

To investigate the underlying mechanics of positional bias, we analyze the discrepancy between the model's internal latent focus and its final predictive output. Figure 3 presents a quantitative analysis averaged over 5 samples where the Ground Truth (GT) is consistently located at Position 4. Specifically, we extract the post-softmax attention weights from the last query token to the visual tokens of each im-

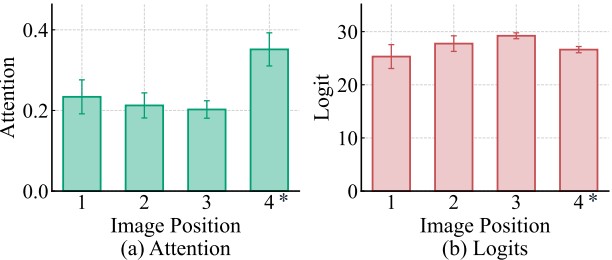

*Figure 3.* A motivating example of the divergence between attention and logit. (a) internal attention weights and (b) final output logits averaged over 5 samples where the Ground Truth (GT) is at position 4 (marked by *). While the model internally attends to the correct position, the final output is hijacked by positional bias, resulting in a spurious peak at position 3. This highlights that correct internal localization does not guarantee unbiased prediction.

age, aggregating them to form a probability distribution. As illustrated in Figure 3(a), the internal attention functions correctly, as the weights consistently peak at the GT index (Pos 4) with low variance. However, a striking reversal occurs in the final projection stage. Figure 3(b) reveals that despite correct internal localization, the final logit distribution is dominated by spurious positional bias, shifting the highest score to Position 3.

We term this phenomenon the *Logit-Attention Divergence*. This observation provides a critical insight: positional bias is likely not a failure of semantic perception (since the model "sees" the right image), but rather a failure of decision calibration. It implies that strong structural priors during the decoding stage override the correct internal features, leading to erroneous predictions despite correct visual grounding.

## 4. Methodology

In this section, we present the Attention-Guided Debiasing framework. We begin in §4.1 by formalizing the multi-image retrieval task and defining the preliminary notations. In §4.2, we propose a counterfactual statistical method to

estimate both the conditional position bias and the static attention prior, requiring only a minimal calibration set. Finally, in §4.3, we describe the inference-time debiasing mechanism, which leverages adaptive layer selection to calculate instance-specific bias from raw logits and visual attention signals.

### 4.1. Preliminaries

We formalize the multi-image retrieval task (Li et al., 2024; Wang et al., 2025a) as follows. Given a text query $q$ and an ordered set of $N$ candidate images $\mathcal{I} = \{v_1, \ldots, v_N\}$, the objective is to predict the index $y^* \in \{1, \ldots, N\}$ of the target image that semantically aligns with $q$. We define the input instance as $x = (q, \mathcal{I})$ and the set of valid candidate answer identifiers as $\mathcal{C} = \{c_1, \ldots, c_N\}$ (e.g., '1', '2', ..., '10', ...), where $c_k$ is the string representation of index $k$. For each candidate, we compute its generation probability $P(c_k|x)$ based on the model's autoregressive distribution. Since $c_k$ may consist of multiple tokens (e.g., '10'), this probability is calculated as the joint probability of the constituent token sequence (detailed in Appendix B).

### 4.2. Prior Probability Statistics.

**Conditional Bias Estimation.** Inspired by the observation in §3.1, we propose a counterfactual statistical approach to estimate position bias using a small calibration dataset $\mathcal{D}_{\text{cal}}$. To decouple positional signals from visual semantics, we construct a symmetrized dataset $\tilde{\mathcal{D}}_{\text{cal}}$ via cyclic permutations. For each instance, we generate $N - 1$ variations by cyclic left-shifting of both the candidate images $\mathcal{I}$ and the ground truth index $y$, ensuring the ground truth traverses every position $k \in \{1, \ldots, N\}$ uniformly. Let $\tilde{\mathcal{D}}_{\text{cal}}^{(i)}$ denote the subset of $\tilde{\mathcal{D}}_{\text{cal}}$ where the ground truth is at position $i$. We compute the empirical conditional probability $\hat{P}_{\text{obs}}(j|i)$, representing the probability of selecting position $j$ given the target is at $i$:

$$\hat{P}_{\text{obs}}(j|i) = \frac{1}{|\tilde{\mathcal{D}}_{\text{cal}}^{(i)}|} \sum_{x \in \tilde{\mathcal{D}}_{\text{cal}}^{(i)}} P(c_j|x). \quad (1)$$

We then model the empirical generation probability as a composition of intrinsic visual evidence and structural priors. Specifically, we postulate that the probability factorizes into a position-dependent bias term $P_{\text{bias}}$ and a target-specific visual signal $P_{\text{vis}}$:

$$P_{\text{obs}}(j \mid i) \propto P_{\text{bias}}(j \mid i) \cdot P_{\text{vis}}(j \mid i), \quad (2)$$

where $P_{\text{bias}}(j \mid i)$ captures the structural tendency to select position $j$ given target $i$. Critically, our $P_{\text{bias}}(j \mid i)$ is conditioned on the ground truth position $i$, enabling it to model the dynamic positional bias patterns observed in §3.1. The term $P_{\text{vis}}(j \mid i)$ represents the idealized visual recognition

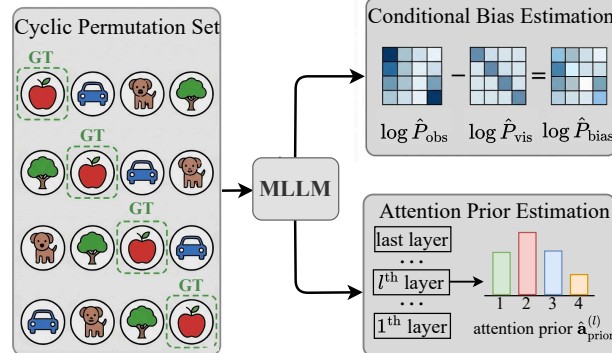

*Figure 4.* Illustration of prior probability statistics (§4.2): we construct a symmetrized calibration set via cyclic permutations to estimate conditional position bias and layer-wise attention priors.

capability, which we parameterize using a scalar confidence gain $\gamma$:

$$P_{\text{vis}}(j \mid i) = \gamma^{\mathbb{I}(j=i)}, \quad \gamma > 1. \quad (3)$$

This formulation implies that visual evidence contributes a multiplicative boost $\gamma$ solely to the ground truth position. To disentangle the bias, we adopt a conservative assumption: the structural bias acts as a distractor and should not inherently favor the ground truth over incorrect positions (i.e., $P_{\text{bias}}(i \mid i) \leq P_{\text{bias}}(j \mid i)$ for $j \neq i$). This assumption reflects the principle that structural priors should not assist correct predictions, allowing us to avoid introducing additional hyperparameters while ensuring $\gamma$ represents the minimum visual signal strength. Under this constraint, the visual gain $\gamma$ is lower-bounded by the likelihood ratio $\frac{P_{\text{obs}}(i|i)}{P_{\text{obs}}(j|i)}$. To estimate the signal strength without over-correction, we define $\hat{\gamma}$ as the maximum observed margin:

$$\hat{\gamma} = \max_{i, j \neq i} \frac{\hat{P}_{\text{obs}}(i \mid i)}{\hat{P}_{\text{obs}}(j \mid i)}. \quad (4)$$

Finally, the pure bias distribution is recovered by removing the estimated visual signal from the empirical observations:

$$\log \hat{P}_{\text{bias}}(j \mid i) = \log \hat{P}_{\text{obs}}(j \mid i) - \log P_{\text{vis}}(j \mid i). \quad (5)$$

This estimated bias $\hat{P}_{\text{bias}}$ serves as the basis for our inference-time correction.

**Attention Prior Estimation.** Drawing on the findings in §3.2 that internal attention aligns better with visual semantics than output logits, we utilize the attention distribution from the last query token $t_q$ as our primary source of visual evidence. For a specific input instance $x$, let $\mathbf{A}^{(l,h)}(x) \in \mathbb{R}^{T \times T}$ denote the post-softmax attention matrix at layer $l$ and head $h$, where $T$ is the sequence length. We compute the instance-specific attention score $a_k^{(l)}(x)$ for the $k$-th image $v_k$ by aggregating the attention weights directed to its visual token span $\mathcal{R}_k$, where $\mathcal{R}_k$ denotes the set of

token indices corresponding to the $k$-th image:

$$a_k^{(l)}(x) = \frac{1}{H} \sum_{h=1}^{H} \sum_{t \in \mathcal{R}_k} \mathbf{A}_{t_q \to t}^{(l,h)}(x). \tag{6}$$

This yields a raw visual evidence vector $\mathbf{a}^{(l)}(x) = [a_1^{(l)}(x), \ldots, a_N^{(l)}(x)]^\top \in \mathbb{R}^N$. However, raw attention scores are also confounded by systematic artifacts such as "attention sinks" (Xiao et al., 2024) and "lost in the middle" (Liu et al., 2024b). To isolate this confounding factor, we estimate a static attention prior $\mathbf{a}_{\text{prior}}^{(l)}$. By averaging the raw attention vectors over the symmetrized calibration set $\tilde{\mathcal{D}}_{\text{cal}}$, we marginalize out the dynamic semantic signals and capture the underlying structural bias:

$$\hat{\mathbf{a}}_{\text{prior}}^{(l)} = \mathbb{E}_{x \sim \tilde{\mathcal{D}}_{\text{cal}}}[\mathbf{a}^{(l)}(x)]. \tag{7}$$

This prior serves as a structural baseline, allowing us to calibrate the visual evidence during inference.

### 4.3. Attention-Guided Debiasing

During inference, we formulate the bias correction as a Bayesian-inspired estimation framework. We first estimate a visual posterior distribution over the candidate images, serving as a proxy for the latent ground truth, and then compute the expected bias under this distribution to rectify the model's prediction.

**Adaptive Visual Evidence Selection.** Visual semantic encoding is often stratified across specific Transformer layers (Raghu et al., 2021). To extract the most informative visual signals, we prioritize layers that exhibit high attention concentrations on image regions. For a test instance $x$, we measure the visual evidence strength of the $l$-th layer by summing the attention probability mass assigned to all candidate images:

$$S^{(l)}(x) = \sum_{k=1}^{N} a_k^{(l)}(x). \tag{8}$$

We select the set of indices $\mathcal{L}^*$ corresponding to the top-$K$ layers with the highest visual evidence strength:

$$\mathcal{L}^* = \underset{l \in \{1, \ldots, L\}}{\text{TopK}} \left( S^{(l)}(x) \right). \tag{9}$$

**Intrinsic Visual Evidence Retrieval.** With the static attention noise quantified as $\mathbf{a}_{\text{prior}}^{(l)}$ in §4.2, our goal is to recover the genuine visual semantics buried within the raw attention. We model the instance-specific attention $a_k^{(l)}(x)$ as a confounded observation that factorizes into an intrinsic visual probability $\tilde{\pi}_k(x)$ and the static structural prior:

$$a_k^{(l)}(x) \propto \tilde{\pi}_k(x) \cdot \mathbf{a}_{\text{prior},k}^{(l)}. \tag{10}$$

Here, $\tilde{\pi}(x) \in \Delta^{N-1}$ represents the model's pure belief about the ground truth location, effectively disentangled from the structural artifacts. To retrieve this intrinsic distribution, we apply the inverse operation in the log-domain, aggregating estimates from the selected informative layers $\mathcal{L}^*$:

$$\log \tilde{\pi}_k(x) \propto \frac{1}{|\mathcal{L}^*|} \sum_{l \in \mathcal{L}^*} \left( \log a_k^{(l)}(x) - \log \mathbf{a}_{\text{prior},k}^{(l)} \right). \tag{11}$$

We further apply temperature scaling with $\tau$ to sharpen this distribution, yielding the final visual posterior

$$\pi(x) = \text{softmax}(\log \tilde{\pi}(x) \cdot \tau). \tag{12}$$

This $\pi$ serves as a purified estimator for the latent ground truth $y$.

**Expected Bias Correction.** Since the true ground truth $y$ is latent during inference, the exact conditional bias term $\hat{P}_{\text{bias}}(\cdot|y)$ cannot be deterministically selected. Instead, we construct an instance-specific dynamic mixture prior by marginalizing the conditional bias profiles over the inferred visual posterior $\pi$. We estimate the expected structural prior for each position $j$ by weighting the pre-computed bias probabilities with the visual belief $\pi$:

$$P_{\text{prior}}(j|x) = \mathbb{E}_{k \sim \pi(x)}[\hat{P}_{\text{bias}}(j|k)] = \sum_{k=1}^{N} \pi_k(x) \hat{P}_{\text{bias}}(j|k). \tag{13}$$

This soft aggregation dynamically adapts the correction strength: when visual evidence is ambiguous (high entropy in $\pi(x)$), the prior converges to a smoothed global bias; when visual evidence is decisive (low entropy), it shifts toward a precise, position-specific bias pattern.

Finally, we recover the intrinsic visual probability $\hat{P}_{\text{vis}}(c_j|x)$ by removing the estimated dynamic prior from the empirical observation $P_{\text{obs}}(c_j|x)$:

$$\log \hat{P}_{\text{vis}}(c_j|x) = \log P_{\text{obs}}(c_j|x) - \log P_{\text{prior}}(j|x). \tag{14}$$

The calibrated distribution $\hat{P}_{\text{vis}}$ represents the model's pure semantic judgment and is used for the final prediction.

## 5. Experiments

### 5.1. Experimental Setup

**Baselines.** We evaluate our method on three representative MLLM backbones: Qwen2.5-VL-3B (Bai et al., 2025), LLaVA-OneVision-8B (An et al., 2025) and InternVL3-8B (Zhu et al., 2025). Unless otherwise specified (e.g., for ensemble methods), we employ greedy decoding for deterministic evaluation. We compare against representative training-free baselines: (1) the vanilla model, utilizing

*Table 1.* Performance on multi-image retrieval ($N = 4$). Following the settings in §5.2, we report Mean Accuracy (Acc), Recall Standard Deviation (RStd), and prediction Consistency (Cons.) for the $N = 4$ case. Bold indicates the best performance among training-free methods.

| Setting | Method | Qwen2.5-VL-3B | | | LLaVA-OneVision-8B | | | InternVL-3-8B | | |
|---------|--------|---------------|---|---|--------------------|---|---|---------------|---|---|
| | | Acc (↑) | RStd (↓) | Cons. (↑) | Acc (↑) | RStd (↓) | Cons. (↑) | Acc (↑) | RStd (↓) | Cons. (↑) |
| Random | Vanilla | 51.70±1.38 | 36.27 | 6.61 | 67.52±1.06 | 25.43 | 19.90 | 69.04±2.09 | 28.38 | 18.50 |
| | Instruction | 48.96±0.97 | 35.80 | 4.20 | 50.50±1.61 | 34.61 | 5.90 | 67.52±1.69 | 26.27 | 19.60 |
| | CoT | 54.16±1.98 | 36.75 | 8.20 | 45.40±1.56 | 34.15 | 3.31 | 77.96±1.46 | 12.45 | 37.70 |
| | PriDe | 63.76±0.88 | 36.24 | 17.70 | 68.00±0.91 | 33.07 | 32.10 | 76.32±0.90 | 18.33 | 35.90 |
| | Self-Consistency | 63.24±1.31 | 20.35 | 15.70 | 41.90±4.55 | 19.09 | 0.00 | 69.20±1.35 | 24.06 | 22.00 |
| | Ours | **94.94±0.52** | **4.20** | **84.30** | **98.66±0.36** | **0.88** | **96.50** | **94.84±0.40** | **5.93** | **80.70** |
| Adversarial | Vanilla | 38.04±1.84 | 32.21 | 4.10 | 49.56±1.20 | 16.62 | 13.90 | 61.00±0.84 | 25.22 | 19.00 |
| | Instruction | 35.92±1.61 | 33.76 | 2.80 | 39.22±1.79 | 28.64 | 3.90 | 57.14±1.18 | 29.95 | 15.20 |
| | CoT | 42.24±0.73 | 18.85 | 3.41 | 36.24±1.06 | 34.04 | 2.71 | 61.08±1.00 | 14.13 | 20.50 |
| | PriDe | 47.80±0.48 | 28.80 | 10.40 | 53.84±0.58 | 31.77 | 20.80 | 64.16±1.29 | 20.65 | 30.40 |
| | Self-Consistency | 42.20±2.09 | 12.96 | 3.70 | 35.70±2.66 | 11.63 | 1.50 | 61.70±1.35 | 21.40 | 20.00 |
| | Ours | **65.72±0.61** | **13.71** | **40.90** | **71.06±1.10** | **10.23** | **51.30** | **76.64±1.45** | **7.89** | **54.10** |

raw logits; (2) prompting strategies, including Instruction Prompting (Zheng et al., 2024) and Chain-of-Thought (CoT) (Wei et al., 2022); and (3) inference-time interventions, such as PriDe (Zheng et al., 2024) and Self-Consistency (Wang et al., 2022).

**Implementation Details.** For the counterfactual bias estimation, we utilize a minimal calibration set of $|\mathcal{D}_{cal}| = 5$ samples, which are distinct from the evaluation set. For a fair comparison, we apply the same calibration budget to the PriDe baseline. In the adaptive layer selection module, we prioritize the top-$K$ layers ($K = 2$) with the highest attention intensity and set the temperature scaling factor $\tau$ to 5.0 for posterior estimation. For the Self-Consistency baseline, we sample 10 reasoning paths with a sampling temperature of 0.7 and adopt majority voting.

### 5.2. Benchmarks

**Datasets.** Following the task formulation in Wang et al. (2025a), we construct a multi-image retrieval evaluation set based on MS-COCO (Lin et al., 2014). To evaluate model performance across varying difficulty levels, we design two distinct evaluation settings based on the candidate sampling strategy:

- **Random Setting:** The candidate images are sampled uniformly from the dataset. This setting establishes a baseline for general visual recognition performance.

- **Adversarial Setting:** To construct hard negative samples, we utilize CLIP (Radford et al., 2021) embeddings to compute cosine similarity between the ground truth and the pool, selecting the nearest neighbors as the negatives.

For each setting, we curate an evaluation set consisting of 1,000 distinct samples. We set the default candidate pool size to $N = 4$ for the main experiments. To assess robustness against larger visual contexts, we evaluate varying

candidate scales in the scalability analysis presented in §5.4.

**Metrics.** Ideally, model predictions should be invariant to the order of candidate images. To evaluate this robustness against input permutations, we apply $T = 5$ random shuffles to each test instance. We report three metrics: (1) Accuracy (Acc): The mean accuracy averaged over the $T$ permutations; (2) Recall Standard Deviation (RStd): The standard deviation of recall rates across the $N$ physical positions, capturing positional bias; and (3) Consistency (Cons.): The percentage of samples where the model selects the same image instance across $T$ permutations, demonstrating permutation invariance.

### 5.3. Overall Results

We report the main results in Table 1 with candidate size $N = 4$. Our attention-guided approach consistently outperforms baselines across different model architectures. Additional experimental results are detailed in Appendix D.

**Recovering Performance under Random Settings.** The results indicate that MLLMs are capable of handling non-adversarial samples when position bias is properly controlled, as reflected by the strong performance across all models under our method. In contrast, severe position bias degrades the performance of the Vanilla model. PriDe partially alleviates this issue via a static global prior, but remains insensitive to instance-level positional bias, resulting in only modest improvement. By explicitly modeling positional effects, our method effectively recovers recognition performance while maintaining low recall standard deviation.

**Robustness against Hard Negatives.** The adversarial setting uses hard negatives to test robustness. When candidates look similar, standard models struggle to distinguish them, leading to a sharp drop in accuracy. Static methods like

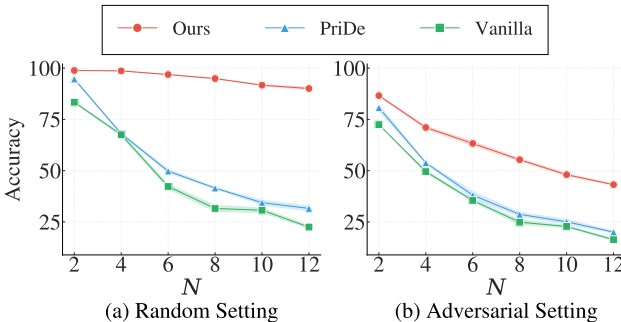

*Figure 5.* Performance comparison across varying candidate pool sizes $N \in \{2, \ldots, 12\}$ on LLaVA-OneVision. While baselines (Blue/Green) drop rapidly due to noise accumulation, our method (Red) demonstrates significantly higher resistance to scaling exhibiting a more gradual decline in accuracy in both settings.

*Table 2.* Computational efficiency comparison on LLaVA-OneVision-8B with varying candidate pool sizes ($N \in \{4, 8, 12\}$). We report the total end-to-end inference latency (s) and the peak GPU memory allocation (MB) on the full test set (1,000 samples).

| Method | Latency (s) ↓ | | | Memory (MB) ↓ | | |
| --- | --- | --- | --- | --- | --- | --- |
| | 4 | 8 | 12 | 4 | 8 | 12 |
| Vanilla | 167.0 | 237.3 | 393.4 | 16,489 | 16,650 | 16,805 |
| PriDe | 187.3 | 258.8 | 552.8 | 16,489 | 16,650 | 16,805 |
| Ours | 197.5 | 266.2 | 568.4 | 16,967 | 18,218 | 20,150 |

PriDe provide limited help because they apply the same correction to every sample, ignoring instance-level confidence variations. In contrast, our method features an adaptive adjustment that applies generic estimates when the model is confused and precise corrections when the visual signal is clear. While our method achieves substantial improvements, the gap between adversarial and random settings reveals an applicability boundary. Position bias correction can recover perception masked by structural priors, but cannot exceed the model's intrinsic discrimination capacity.

**Prediction Stability.** Accuracy alone can be misleading. We observe that Vanilla models show low consistency across all backbones, even when their accuracy appears reasonable, indicating that many correct predictions are unstable under input permutations. In contrast, our method achieves high consistency, suggesting that predictions are driven by visual evidence instead of positional artifacts. Moreover, prompting strategies do not alleviate this issue and Instruction even performs worse than Vanilla in several cases. This observation implies that position bias arises from low-level model behavior and cannot be corrected by prompting alone, motivating our inference-time logit calibration.

### 5.4. Impact of Candidate Pool Size

To evaluate how the number of candidates affects model performance, we expand the candidate pool size $N$ from 2 to 12, which increases sequence length and introduces more visual distractors, making the selection task more difficult.

**Robustness to Context Scaling.** As shown in Figure 5, all methods decline in accuracy as $N$ increases due to rising task complexity. However, the rate of degradation differs between approaches. The Vanilla and PriDe baselines suffer rapid decay, showing limited ability to distinguish similar images in long contexts. In contrast, our method maintains higher accuracy with a more gradual slope. These results provide empirical evidence for the Logit-Attention divergence discussed in §3.2. As visual context expands, po-

sitional priors increasingly dominate the generation process, leading to corrupted output logits. By utilizing instance-specific attention signals, our approach bypasses these biased predictions. This demonstrates that the internal visual perception of MLLMs is more resilient to context scaling than their final autoregressive distributions, and this latent capability can be recovered without retraining.

**Computational Efficiency.** As detailed in Table 2, we evaluate scalability in terms of total latency and memory. A key advantage is that our method operates within a single forward pass, avoiding ensemble costs. The latency overhead compared to Vanilla primarily stems from the fixed setup cost for counterfactual bias estimation ($N \times |\mathcal{D}_{cal}|$), which can be managed by reducing $|\mathcal{D}_{cal}|$ as $N$ increases. Moreover, multi-token scenarios (e.g., $N = 12$) incur additional latency from exact candidate scoring, as calibration methods must perform full forward passes to compute joint probabilities for all multi-token options. Notably, the attention extraction in our method introduces negligible latency because it involves only $\mathcal{O}(N^2)$ vector operations. In terms of memory, our method incurs higher usage than the logit-only baselines due to caching intermediate attention maps, yet remains well within the capacity of consumer-grade GPUs.

### 5.5. Generalization

To assess whether the estimated bias correction is structurally invariant or dataset-dependent, we evaluate our method under cross-domain and cross-difficulty shifts. We compare our approach against the Vanilla baseline and PriDe to test the generalization of learned priors.

**Cross-Domain Generalization.** This scenario examines whether positional bias is an architectural decoder artifact or tied to specific dataset semantics. As summarized in Table 3, when transferring the calibration between the Flickr8k (Hodosh et al., 2013) and MS-COCO (Lin et al., 2014) datasets, our method demonstrates high stability. For instance, calibrating on Flickr8k and testing on MS-COCO yields 97.78% accuracy and 93.40% consistency. Compared to PriDe, our attention-guided mechanism identifies dataset-independent structural artifacts more effectively, validating that our framework captures intrinsic semantic structures

*Table 3.* We compare robustness against cross-domain and cross-difficulty on LLaVA-OneVision, where $A \rightarrow B$ means calibrating on $A$ and testing on $B$. Abbreviations: C(MSCOCO), F(Flickr8k), R(Random), and A(Adversarial). Vanilla denotes the baseline without calibration. Bold indicates the best results.

| Setting | Method | Acc (↑) | RStd (↓) | Cons. (↑) |
|---|---|---|---|---|
| $F(R) \rightarrow C(R)$ | Vanilla | 67.52±1.06 | 25.43 | 19.90 |
| | PriDe | 62.54±1.31 | 41.79 | 23.70 |
| | Ours | **97.78±0.40** | **1.60** | **93.40** |
| $C(R) \rightarrow F(R)$ | Vanilla | 57.58±0.72 | 28.97 | 9.40 |
| | PriDe | 64.36±1.40 | 34.45 | 28.00 |
| | Ours | **95.92±0.16** | **4.16** | **88.10** |
| $C(R) \rightarrow C(A)$ | Vanilla | 49.56±1.20 | 16.62 | 13.90 |
| | PriDe | 43.00±0.94 | 41.73 | 13.90 |
| | Ours | **65.94±0.57** | **17.51** | **44.60** |
| $C(A) \rightarrow C(R)$ | Vanilla | 67.52±1.06 | 25.43 | 19.90 |
| | PriDe | 68.88±0.83 | 33.09 | 24.40 |
| | Ours | **96.98±0.22** | **2.94** | **90.00** |

that generalize across distinct visual domains.

**Cross-Difficulty Generalization.** We evaluate generalization between difficulty levels by transferring between random and adversarial splits. In the challenging Random-to-Adversarial transfer setting, where adversarial samples increase visual uncertainty and exacerbate positional priors, our approach maintains a robust 65.94% accuracy, significantly outperforming both Vanilla and PriDe baselines. Similarly, in the Adversarial-to-Random scenario, our method retains a high accuracy of 96.98%. These results demonstrate that the structural bias captured during calibration generalizes across varying entropy levels, achieving stable performance without requiring task-specific recalibration.

### 5.6. Ablation Study

**Sample Efficiency of Bias Estimation.** We analyze the sensitivity of our method to the calibration set size $|\mathcal{D}_{cal}|$. As shown in Figure 6(a), performance reaches a plateau with as few as 5 samples. This efficiency stems from the synergy between the structural prior and the dynamic attention mechanism. While the bias matrix provides a foundational estimate of positional preferences, the correction is instance-specifically weighted by the attention-guided posterior $\pi$. Since internal attention often retains valid semantic alignment even when output logits are corrupted, this robust signal effectively compensates for calibration data sparsity, allowing our framework to maintain high accuracy over static baselines even under low-data constraints.

**Posterior Sharpening.** We examine the temperature parameter $\tau$ for modulating the entropy of the attention-guided posterior $\pi$. Sharpening is essential because raw attention weights are often near-uniform. Such flat distributions fail to provide the contrast required to steer bias correction ef-

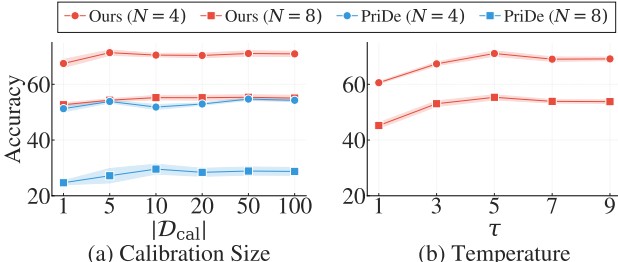

(a) Calibration Size    (b) Temperature

*Figure 6.* Ablation on calibration efficiency and posterior sharpening on LLaVA-OneVision under the adversarial setting. (a) Impact of calibration set size $|\mathcal{D}_{cal}|$ on accuracy. (b) Impact of the temperature parameter $\tau$ on accuracy. Shaded regions denote the standard deviation across 5 random shuffles.

fectively. As shown in Figure 6(b), accuracy improves as $\tau$ increases contrast between candidates. Performance peaks at $\tau = 5$ and stabilizes thereafter, suggesting that additional sharpening yields limited further gains.

**Attention-Guided Mechanism.** We evaluate the contribution of each component in Table 4 under the adversarial setting, where high visual similarity among candidates makes the contribution of each design choice more apparent. First, directly using attention weights for prediction (Attn Readout) yields only 42.36% accuracy, which improves to 64.38% after applying the static attention prior, confirming that raw attention is also confounded by structural artifacts. Our full method further improves to 71.06% by incorporating conditional bias calibration, demonstrating the complementary benefits of both calibration stages. For layer selection, replacing dynamic selection with averaging all layers causes a notable performance drop, confirming that visual evidence is concentrated in specific layers. Regarding $K$, our framework demonstrates robustness across $K \in \{1, 2, 4, 8\}$, with $K = 2$ achieving optimal balance while larger values introduce noise from non-visual layers.

*Table 4.* Ablation on attention-guided mechanisms on LLaVA-OneVision (adversarial setting, $N = 4$). We evaluate attention readout, layer selection strategy, static attention prior, and the sensitivity of hyper-parameter $K$. Bold indicates the best results.

| Strategy | Prior | $K$ | Acc (↑) | RStd (↓) | Cons. (↑) |
|---|---|---|---|---|---|
| Attn Readout | – | 2 | 42.36±0.92 | 37.65 | 9.50 |
| Attn Readout | ✓ | 2 | 64.38±1.04 | 10.17 | 45.30 |
| All Layers | ✓ | – | 61.50±0.84 | 15.93 | 34.10 |
| Dynamic | – | 2 | 58.90±0.17 | 26.65 | 31.70 |
| Dynamic | ✓ | 1 | 68.16±0.98 | 11.68 | 46.30 |
| Dynamic | ✓ | 2 | **71.06±1.10** | **10.23** | **51.30** |
| Dynamic | ✓ | 4 | 68.38±1.16 | 12.33 | 48.30 |
| Dynamic | ✓ | 8 | 66.66±1.21 | 12.41 | 44.90 |

*Table 5.* Generalization to multi-image VQA on six MMIU (Meng et al., 2025) tasks with LLaVA-OneVision, using identical hyperparameters as the retrieval setting. PurAttn (purified attention alone, without bias correction) directly probes Logit-Attention Divergence. ΔRStd is the change relative to Vanilla.

| Task | Vanilla (↑) | PurAttn (↑) | Ours (↑) | ΔRStd (↓) |
|---|---|---|---|---|
| forensic_forgerynet | 35.0 | 86.0 | 87.4 | −39.7 |
| forensic_blink | 24.4 | 27.9 | 30.9 | −11.8 |
| emotion_expw | 28.4 | 30.8 | 31.8 | −15.4 |
| emotion_findingemo | 24.3 | 25.8 | 26.9 | −6.0 |
| text2image_retrieval | 24.3 | 24.6 | 25.2 | −11.6 |
| visual_quality | 49.8 | 49.0 | 53.0 | −35.0 |

### 5.7. Generalization to Multi-Image VQA

We further evaluate on six multi-image VQA tasks from MMIU (Meng et al., 2025), covering forensic detection (forensic_forgerynet, forensic_blink), emotion recognition (emotion_expw, emotion_findingemo), text-to-image retrieval (text2image_retrieval) and visual quality assessment (visual_quality). These tasks share the discrete-choice structure with our retrieval setting, where answer candidates are expressed as ordinal references (e.g., "the first image", "the second image"), but differ substantially in visual semantics, spanning low-level forgery detection to high-level emotion recognition, which makes them a natural testbed for out-of-domain generalization. We apply identical hyperparameters without task-specific tuning. To directly probe the divergence phenomenon, we additionally report Purified Attention (PurAttn), i.e., the attention posterior $\pi$ (Eq. 12) used alone without the conditional bias correction.

As shown in Table 5, PurAttn matches or exceeds Vanilla on 5 of 6 tasks. The most striking case is forensic_forgerynet, where PurAttn reaches $86.0\%$ against Vanilla's $35.0\%$, a gap that is difficult to attribute to visual perception alone. This raises a broader concern: on certain MLLM benchmarks, reported low accuracy may reflect positional bias rather than genuine capability limitations, as the model already attends to the correct answer internally but fails to express it in the output. Our full method further improves accuracy on every task and reduces RStd consistently. Gains are more modest on visually ambiguous tasks, where the attention signal itself is less reliable, suggesting that the effectiveness of our framework is bounded by the semantic faithfulness of the model's internal attention. How to further improve the reliability of internal attention under high visual ambiguity remains an open question.

### 6. Conclusion

We investigate position bias arising in in-context cross-modal retrieval and identify Logit-Attention Divergence, where correct visual localization is overridden by structural decoding priors. Based on this insight, we propose a training-free, attention-guided debiasing framework that leverages intrinsic visual signals for recalibration. Our approach restores permutation invariance and achieves state-of-the-art performance with minimal computational overhead, highlighting the unexploited potential of internal attention mechanisms for enhancing MLLM robustness in multi-image contexts.

### Acknowledgments

This work is supported by New Generation Artificial Intelligence-National Science and Technology Major Project (No. 2025ZD0122901). This work is also supported by National Science Foundation of China (No. 62572313, No. 62106139).

### Impact Statement

This paper presents work whose goal is to advance the field of Machine Learning. There are many potential societal consequences of our work, none which we feel must be specifically highlighted here.

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

## A. Limitations

We acknowledge specific limitations regarding the operational requirements and scope of our framework. Crucially, the reliance on extracting internal attention weights necessitates white-box access, making the method inapplicable to closed-source APIs where intermediate activations are unavailable. Furthermore, our method relies on the semantic faithfulness of the model's attention, which is inherently bounded by the visual encoder's capabilities; yet, our consistent performance on adversarial datasets demonstrates that this internal signal remains robust even when the output generation is compromised. Additionally, the current formulation is strictly tailored to discriminative retrieval over discrete candidate sets and does not straightforwardly extend to open-ended generative tasks.

## B. Formal Derivation of Joint Candidate Probabilities

In multi-image retrieval tasks, the model is required to predict a discrete index $y \in \{1, \ldots, N\}$ corresponding to the target image. We represent each candidate answer as a string identifier $c_k$ (e.g., "1", "2", ..., "10", ...). Critically, when $N > 9$, some identifiers consist of multiple tokens, requiring careful handling of the autoregressive generation process. Given an input instance $x = (q, \mathcal{I})$ where $q$ is the text query and $\mathcal{I} = \{v_1, \ldots, v_N\}$ is the ordered set of candidate images, the model generates the answer identifier $c_y$ autoregressively. We denote the tokenization of a candidate identifier as $c_k = \langle t_k^{(1)}, t_k^{(2)}, \ldots, t_k^{(m_k)} \rangle$, where $m_k$ is the sequence length. The generation probability of candidate $c_k$ is defined as the joint probability of its constituent token sequence:

$$\log P(c_k \mid x) = \sum_{j=1}^{m_k} \log P\left(t_k^{(j)} \mid x, t_k^{(1)}, \ldots, t_k^{(j-1)}\right) \tag{15}$$

For single-token candidates ($k \leq 9$), this includes an end-of-sequence (EOS) token: $\log P(c_k \mid x) = \log P(t_k \mid x) + \log P(\text{EOS} \mid x, t_k)$. For multi-token candidates ($k \geq 10$), the probability decomposes into prefix and suffix components. To ensure valid probability distributions over the candidate set $\mathcal{C}$, we apply restricted softmax normalization at each generation step. For the $j$-th token position, we normalize the raw logit $\ell_k^{(j)}$ over the set of valid tokens $\mathcal{T}_j$ at that position:

$$\log P(t_k^{(j)} \mid x, t_k^{(1)}, \ldots, t_k^{(j-1)}) = \ell_k^{(j)} - \log \sum_{t \in \mathcal{T}_j} \exp(\ell_t^{(j)}) \tag{16}$$

where $\mathcal{T}_1 = \{t_1^{(1)}, \ldots, t_N^{(1)}\}$ denotes all possible first tokens, and $\mathcal{T}_j$ for $j > 1$ denotes valid continuation tokens given the prefix (including EOS where appropriate). This restricted normalization ensures that $\sum_{k=1}^{N} P(c_k \mid x) = 1$, allowing for fair comparison between candidates of varying token lengths. In practice, this requires at most two forward passes: one for first tokens and grouped passes for each unique prefix.

## C. Construction of the Adversarial Benchmark

To evaluate MLLMs under high visual confusion, we constructed an adversarial benchmark using CLIP-based (Radford et al., 2021) hard negative mining on MS-COCO (Lin et al., 2014) and Flickr8k (Hodosh et al., 2013). Unlike random sampling, this setting selects distractors that are visually similar to the target but semantically distinct. For each target image $v_{anc}$, we extract $\ell_2$-normalized CLIP (ViT-L/14) embeddings for both images and captions. We then select $N - 1$ distractors $v_{neg}$ from the pool $\mathcal{D}$ that satisfy the following constraints: (1) We first construct a filtered candidate set $\mathcal{D}_{\text{filtered}}$ by excluding images with category sets identical to the anchor and enforcing a textual cosine similarity threshold $S_{\text{txt}}(e_{\text{anc}}^{\text{txt}}, e_{\text{neg}}^{\text{txt}}) < 0.9$ to ensure a unique and unambiguous ground truth; (2) hard distractors are then selected by maximizing the visual cosine similarity to the anchor within this filtered pool: $v_{\text{neg}}^* = \arg\max_{v \in \mathcal{D}_{\text{filtered}}} \cos(e_{\text{anc}}^{\text{vis}}, e_v^{\text{vis}})$.

This strategy enhances benchmark difficulty by introducing high visual ambiguity. As shown in Figure 7, distractors often share similar backgrounds or lighting with the ground truth, but they remain semantically inconsistent with the query. For instance, in Figure 7b, while multiple candidates feature motorcycles, only the ground truth aligns with specific details like a "red motorcycle" and a "man in a black jacket." This approach ensures that MLLMs must employ fine-grained reasoning rather than simple keyword matching.

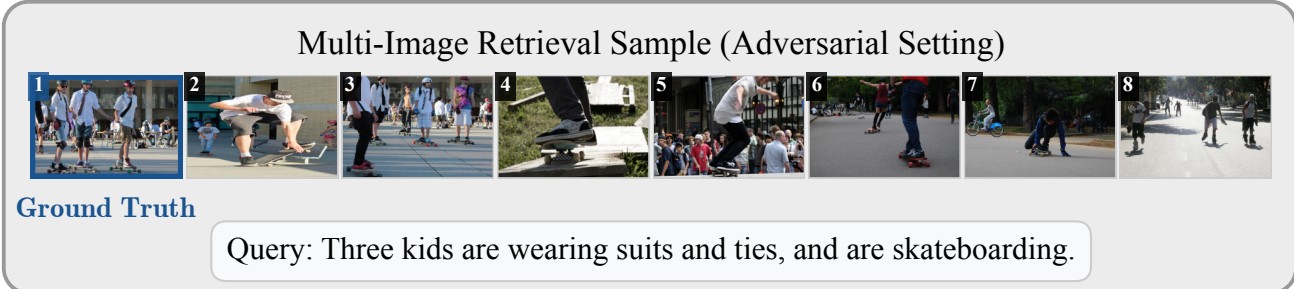

*(a)* Query: "Three kids are wearing suits and ties, and are skateboarding."

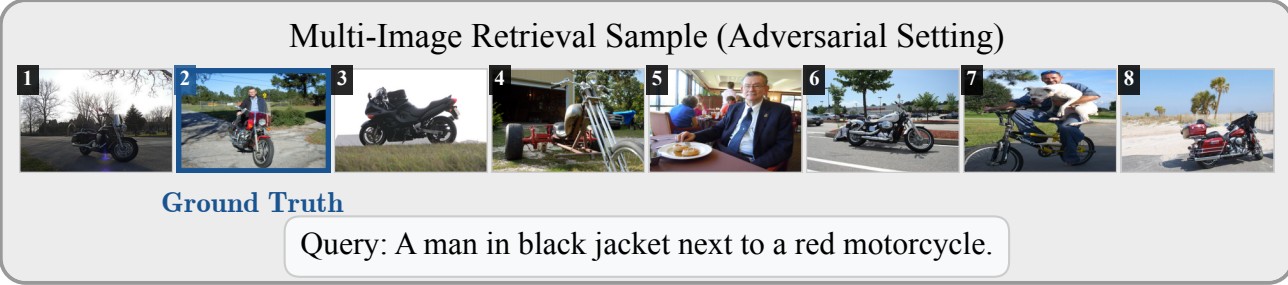

*(b)* Query: "A man in black jacket next to a red motorcycle."

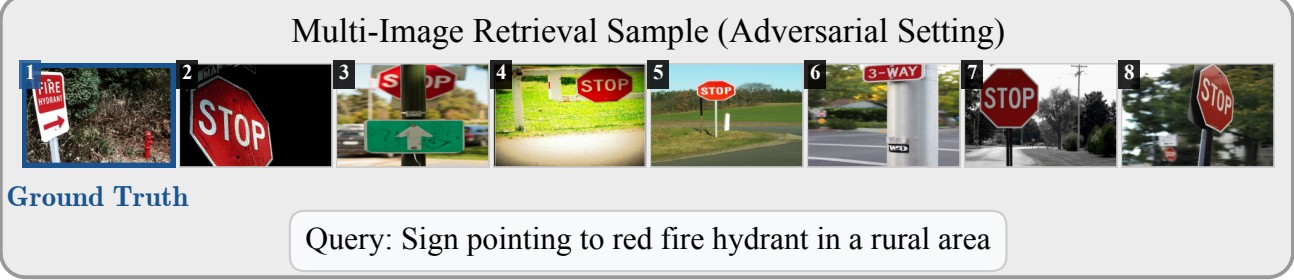

*(c)* Query: "Sign pointing to red fire hydrant in a rural area."

*Figure 7.* Examples of multi-image retrieval samples under the adversarial setting.

## D. Supplementary Experimental Results

### D.1. Performance on Larger Candidate Pools

To further evaluate the scalability of our approach, we report complete results for $N = 8$ candidates in Table 6. Compared to the $N = 4$ setting presented in the main paper (Table 1), the expanded candidate pool introduces significantly greater complexity: the probability of random guessing drops from 25% to 12.5%, and the accumulation of visual distractors intensifies both positional bias and semantic ambiguity.

As shown in Table 6, our method maintains strong performance even under this increased difficulty. In the Random setting, we consistently achieve over 90% accuracy across Qwen2.5-VL-3B (Bai et al., 2025), LLaVA-OneVision-8B (An et al., 2025), and InternVL3-8B (Zhu et al., 2025), demonstrating consistent effectiveness across architectures. The substantial improvements in Consistency metrics confirm that our attention-guided calibration successfully restores permutation invariance despite the larger context. In the more challenging adversarial setting, where hard negatives exhibit high visual similarity to the target, our method achieves 52.96%, 55.34%, and 65.00% accuracy respectively, substantially outperforming baselines such as PriDe (Zheng et al., 2024), while maintaining low recall standard deviation (RStd). These results validate that our framework scales gracefully to longer visual contexts while maintaining robustness against both random and adversarial candidate distributions.

*Table 6.* Performance on multi-image retrieval ($N = 8$). Following the settings in §5.2, we report Mean Accuracy (Acc), Recall Standard Deviation (RStd), and prediction Consistency (Cons.) for the $N = 8$ case. Bold indicates the best performance among training-free methods.

| Setting | Method | Qwen2.5-VL-3B | | | LLaVA-OneVision-8B | | | InternVL-3 | | |
|---|---|---|---|---|---|---|---|---|---|---|
| | | Acc (↑) | RStd (↓) | Cons. (↑) | Acc (↑) | RStd (↓) | Cons. (↑) | Acc (↑) | RStd (↓) | Cons. (↑) |
| Random | Vanilla | 28.68±1.55 | 28.86 | 1.20 | 31.56±2.01 | 38.82 | 0.10 | 51.00±1.70 | 31.27 | 5.80 |
| | Instruction | 28.86±1.74 | 28.24 | 1.30 | 26.72±2.59 | 35.49 | 0.20 | 45.44±1.57 | 29.28 | 3.10 |
| | CoT | 31.94±2.52 | 29.39 | 2.91 | 30.50±1.90 | 37.19 | 1.30 | 31.84±1.21 | 12.50 | 0.70 |
| | PriDe | 34.62±1.80 | 30.40 | 2.30 | 41.52±0.63 | 34.51 | 1.00 | 54.52±2.13 | 29.87 | 4.30 |
| | Ours | **93.10±1.02** | **3.68** | **76.80** | **94.92±0.40** | **3.33** | **84.10** | **90.54±0.94** | **13.89** | **66.70** |
| Adversarial | Vanilla | 19.58±1.09 | 28.48 | 1.20 | 24.88±1.63 | 32.58 | 0.30 | 37.00±1.88 | 29.86 | 3.30 |
| | Instruction | 19.02±0.59 | 30.17 | 0.80 | 22.02±1.36 | 33.49 | 0.10 | 34.06±1.35 | 26.35 | 2.60 |
| | CoT | 23.34±1.51 | 23.15 | 1.81 | 22.66±0.87 | 31.02 | 0.40 | 20.34±0.97 | 13.61 | 0.10 |
| | PriDe | 25.30±0.87 | 28.52 | 2.30 | 28.70±1.32 | 28.37 | 0.20 | 38.22±2.02 | 30.25 | 1.40 |
| | Ours | **52.96±1.53** | **13.36** | **28.00** | **55.34±1.04** | **11.10** | **28.10** | **65.00±0.66** | **7.79** | **35.20** |

## D.2. Robustness to Diverse Candidate Identifiers

Our main experiments primarily utilize numeric identifiers (e.g., `"1"`, `"2"`, ...) as candidate labels. To verify that our method generalizes beyond this specific tokenization scheme, we evaluate performance using four alternative identifier formats: uppercase alphabetic characters (`"A"`, `"B"`, ...), lowercase alphabetic characters (`"a"`, `"b"`, ...), Roman numerals (`"I"`, `"II"`, ...), and number words (`"first"`, `"second"`, ...). As visualized in Figure 8, the Vanilla model exhibits severe position bias across all formats. For instance, in the number words setting, the accuracy for the "Third" position collapses to a negligible $0.5\%$, while the first candidate dominates with a spuriously high selection rate of $43.7\%$. In contrast, our method successfully restores the diagonal retrieval structure, consistently achieving over $96\%$ accuracy regardless of the identifier type. These results confirm that the Logit-Attention Divergence is a fundamental phenomenon rooted in visual grounding mechanisms, rather than an artifact of specific tokenizer behaviors.

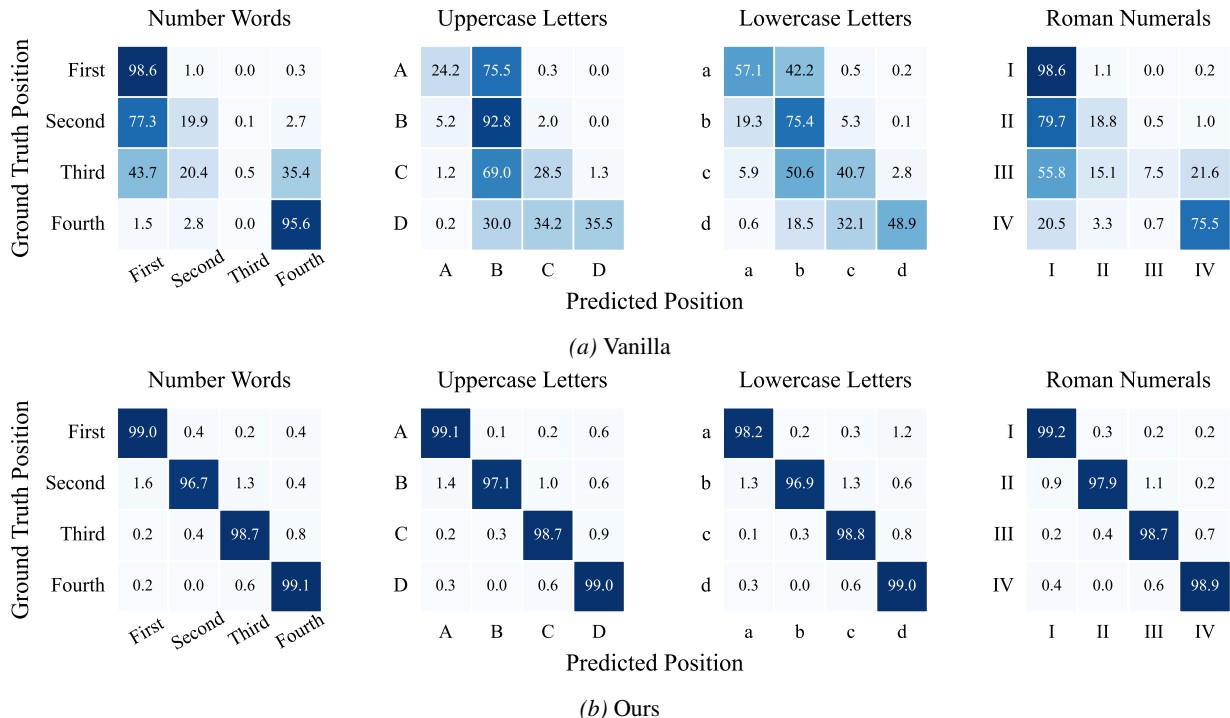

*Figure 8.* We evaluate LLaVA-OneVision-8B ($N = 4$) under the Random setting using four identifier formats. While the Vanilla model (top) shows severe bias, our method (bottom) consistently restores accurate retrieval across all tokenization schemes.

## D.3. Systematic Validation of Logit-Attention Divergence

Figure 3 showed the divergence at a single ground-truth position averaged over five samples. Table 7 extends the comparison to the full evaluation set with LLaVA-OneVision across all GT positions, both candidate sizes, and both difficulty settings. We compare Vanilla logits, Raw Attention argmax (no correction), Purified Attention (the posterior $\pi$ in Eq. 12) and our full method.

Purified Attention exceeds Vanilla in every setting, and the gap widens from 28.6 points at Random $N=4$ to 59.6 points at Random $N=8$, indicating that the divergence becomes more pronounced as the input grows. The position-wise breakdown clarifies the failure mode of Vanilla. At $N=8$, per-position recall under Vanilla ranges from $0.00\%$ at Position 4 to $98.90\%$ at Position 7, while Purified Attention falls in a tighter band of $87\%$ to $96\%$ across positions. The visual signal itself is roughly position invariant, and what gets distorted is the projection from attention to output logits. Raw Attention is actually worse than Vanilla in the adversarial $N=4$ setting ($42.36\%$ versus $49.56\%$), because the attention sink at boundary positions acts as a strong positional signal of its own. Removing the static prior recovers 22.0 points and crosses Vanilla, consistent with the sink being a stable structural artifact rather than instance dependent noise.

*Table 7.* Systematic validation of Logit-Attention Divergence on LLaVA-OneVision across the full evaluation set. Purified Attention beats Vanilla logits in every setting. Best in bold.

| Setting | Vanilla ↑ | Raw Attn ↑ | Purified Attn ↑ | Ours ↑ |
|---|---|---|---|---|
| Random $N=4$ | 67.52 | 91.08 | 96.14 | **98.66** |
| Random $N=8$ | 31.56 | 80.78 | 91.16 | **94.92** |
| Adv $N=4$ | 49.56 | 42.36 | 64.38 | **71.06** |
| Adv $N=8$ | 24.88 | 32.84 | 48.48 | **55.34** |

## D.4. Comparison with SoFA and Permutation Averaging

Two additional baselines on LLaVA-OneVision help locate our method in the broader design space. SoFA (Tian et al., 2025) replaces causal attention with bidirectional attention at inference, a structural intervention that requires no calibration data. Permutation Averaging runs $N$ forward passes with cyclically shifted candidate lists, maps the resulting logits back to the original image identities, and then averages them, which reduces order-dependent bias by symmetrizing predictions over candidate positions without any parametric assumption. Its accuracy serves as an empirical reference point for any factorization-based correction. Results are reported in Table 8.

SoFA falls behind Vanilla in both N=4 settings but marginally surpasses it at N=8, since the bidirectional mask used at inference is not the mask the model was trained under, and the resulting representation shift outweighs the bias mitigation when the candidate pool is small. Permutation Averaging is very strong at $N=4$, and our single-pass method tracks it within 0.4 points at Random $N=4$ and 3.3 points at Adversarial $N=4$. At Random $N=8$ our method comes in 2.1 points ahead ($94.92\%$ versus $92.86\%$), while Adversarial $N=8$ is the only setting where Permutation Averaging is meaningfully better ($60.16\%$ versus $55.34\%$). Permutation Averaging multiplies inference time by $N$, which makes it least practical in exactly the regime where positional bias matters most. The close agreement at single-pass cost also serves as a sanity check on our modeling: a method that erases positional bias by symmetry and a method that erases it via a five-sample calibration produce nearly the same predictions. This is the pattern one would expect if the multiplicative factorization in Eq. (2), equivalently additive after taking logs, is approximately correct.

*Table 8.* Comparison with SoFA (Tian et al., 2025) and Permutation Averaging on LLaVA-OneVision. Permutation Averaging incurs $N\times$ inference cost while our method achieves comparable or superior accuracy in a single forward pass.

| Setting | Vanilla ↑ | SoFA ↑ | Perm. Avg. ↑ ($N\times$forward) | Ours ↑ |
|---|---|---|---|---|
| Random $N=4$ | 67.52 | 59.70 | 99.04 | 98.66 |
| Random $N=8$ | 31.56 | 32.16 | 92.86 | 94.92 |
| Adv $N=4$ | 49.56 | 43.26 | 74.32 | 71.06 |
| Adv $N=8$ | 24.88 | 25.34 | 60.16 | 55.34 |

# E. Prompt Templates

To ensure the reproducibility of our results, we provide the full text of the prompt templates used across all experimental settings. For each method, the $N$ target images $\{v_1, \ldots, v_N\}$ are presented sequentially as the visual input, followed by the specific text prompt detailed below. Note that the calibration-based methods, including PriDe and our proposed Attention-Guided Debiasing, utilize the same prompt as the Vanilla baseline to isolate the effects of logit calibration from prompt-level interventions.

*Table 9.* Consolidated prompt templates for baseline and proposed methods

---

**Vanilla Model Prompt**

---

*Input Structure:* `<Image 1> ...  <Image N>`

Given $N$ images indexed from 1 to $N$, identify the image that best matches the provided caption. Respond with the index number only and nothing else.

Caption: {caption}

Answer:

---

**Instruction Prompt**

---

*Input Structure:* `<Image 1> ...  <Image N>`

Given $N$ images indexed from 1 to $N$, identify the image that best matches the provided caption. Please note that the provided images have been randomly shuffled, so it is essential to consider them fairly and without bias. Analyze the visual content objectively and do not be influenced by the order of the images. Respond with the index number only and nothing else.

Caption: {caption}

Answer:

---

**Chain-of-Thought (CoT) Prompt**

---

*Input Structure:* `<Image 1> ...  <Image N>`

Given $N$ images indexed from 1 to $N$, identify the image that best matches the provided caption. Please think step by step. First, write your reasoning inside <think> and </think> tags. Then, after the closing </think> tag, output only the final answer index (a single integer from 1 to $N$).

Caption: {caption}

Answer: <think>

---

**Self-Consistency Prompt**

---

*Input Structure:* `<Image 1> ...  <Image N>`

Given $N$ images indexed from 1 to $N$, identify the image that best matches the provided caption. Think step by step to analyze the content of the images. After your reasoning, provide the final answer in the exact format: "The answer is <index>".

Caption: {caption}

Answer:

---

