# OpenReview forum: "Logit-Attention Divergence: Mitigating Position Bias in Multi-Image Retrieval via Attention-Guided Calibration"
_ICML.cc/2026/Conference — ICML 2026 regular_

### Official Review · Reviewer_P5Tm · 2026-02-26

**Soundness:** 3
**Presentation:** 3
**Significance:** 2
**Originality:** 3
**Overall Recommendation:** 4
**Confidence:** 4

**Summary:**

This paper studies position bias in Multimodal Large Language Models (MLLMs) for multi-image retrieval tasks, where models prefer certain input positions regardless of semantic relevance.
The authors identify a phenomenon called Logit-Attention Divergence: while output logits are heavily corrupted by positional priors, internal attention maps remain semantically aligned with the correct target.
Based on this insight, they propose a training-free, attention-guided debiasing framework that estimates conditional position bias via counterfactual cyclic permutations on a small calibration set and uses purified attention signals as a visual posterior to construct instance-specific bias corrections at inference time.
Experiments on MS-COCO-based benchmarks with three MLLM backbones show substantial improvements in accuracy, permutation invariance, and prediction consistency across both random and adversarial settings with varying candidate pool sizes.

**Compliance With Llm Reviewing Policy:**

Affirmed.

**Final Justification:**

The authors conducted detailed experiments and response, which addressed my concerns. Based on the overall quality of the manuscript and the authors' response, I'm inclined to maintain the current rating.

**Key Questions For Authors:**

Q1. The Logit-Attention Divergence is demonstrated with only 5 averaged samples at a single GT position (Figure 3). Can you provide a systematic quantitative analysis showing how frequently attention correctly identifies the ground truth across all positions, candidate sizes, and difficulty settings? In particular, what is the attention-based "accuracy" (i.e., argmax of attention = GT) before and after prior removal?

Q2. The factorization in Eq. 2 assumes independence between positional bias and visual signal. Have you tested whether this assumption holds empirically, for instance by checking whether the residuals after bias removal correlate with position? If the interaction is non-multiplicative, how sensitive is the framework to this model misspecification?

Q3. Why is SoFA excluded from experimental comparisons despite being discussed as a relevant baseline in related work? Including this comparison, or explaining why it was omitted, would strengthen the evaluation. Similarly, have you considered comparing against simple permutation-averaging (running inference on all N cyclic permutations and averaging logits)?

Q4. Can you provide evidence that the Logit-Attention Divergence phenomenon extends beyond the multi-image retrieval setting? For instance, does it also manifest in multi-image VQA or other multi-image reasoning tasks where position bias has been documented?

**Limitations:**

Yes.

**Strengths And Weaknesses:**

**Strengths**

S1. Compelling and well-evidenced core observation.
The Logit-Attention Divergence finding, that models "look right but speak wrong," is clearly demonstrated through Figure 3 and supported by systematic analysis. The conditional nature of position bias (Figure 2), where bias shifts with ground truth location, provides a convincing critique of static calibration methods like PriDe [1] and motivates the proposed conditional approach.

S2. Good empirical results with thorough evaluation design.
The improvements are large and consistent: over 40% accuracy gains across three architectures. The evaluation design is notably rigorous, with permutation-based testing (T=5 shuffles), three complementary metrics (accuracy, recall std, consistency), scalability analysis up to N=12, cross-domain/cross-difficulty generalization (Table 3), and robustness to diverse identifier formats (Figure 8). The adversarial benchmark using CLIP-based hard negatives adds meaningful difficulty.

S3. Practical and efficient framework.
The method requires only 5 calibration samples, operates within a single forward pass, and introduces negligible computational overhead (Table 2). The ablation study confirms robustness to calibration size and hyperparameters.


**Weaknesses**

W1.Task scope limits significance.
The method is strictly designed for discriminative multi-image retrieval with discrete candidate sets, as acknowledged in the limitations. It is unclear whether the Logit-Attention Divergence phenomenon or the proposed solution extends to broader MLLM tasks where position bias also matters, such as multi-image VQA, interleaved image-text reasoning, or ranking tasks.

W2. The attention faithfulness assumption lacks rigorous validation.
The entire framework hinges on the claim that internal attention remains semantically faithful despite biased logits.
However, this is demonstrated on only 5 averaged samples in Figure 3 for a single GT position. Prior work has questioned whether attention weights reliably reflect model reasoning [2, 3]. A more rigorous validation would include quantitative attention-accuracy correlation across all positions and candidate sizes, and analysis of when this assumption breaks down, particularly in the adversarial setting where performance drops significantly.

W3. The factorization model (Eq. 2) relies on strong assumptions.
The decomposition of observed probabilities into independent bias and visual signal terms (Eq. 2), the parameterization of visual signal as a single scalar boost γ (Eq. 3), and the constraint that bias should not favor the ground truth are all strong modeling assumptions. The paper does not validate whether these assumptions hold empirically or analyze sensitivity to violations. If the true bias-signal interaction is more complex, the estimated corrections may be suboptimal.

-----
Refs:
[1] Zheng, C., et al. (2023). Large Language Models Are Not Robust Multiple Choice Selectors. ICLR 2024. \
[2] Jain, S., & Wallace, B.C. (2019). Attention is not Explanation. NAACL 2019. \
[3] Wiegreffe, S., & Pinter, Y. (2019). Attention is not not Explanation. EMNLP 2019.

---

> ### Author Rebuttal · Authors · 2026-03-29
>
> We thank you for recognizing our compelling core observation, rigorous evaluation design, and practical framework.
>
> > **W1. Task scope limits significance. (Also addresses Q4)**
>
> We select 6 multi-image VQA tasks from MMIU [1] whose answers are grounded in candidate images (e.g., "the first image"), matching our framework's assumptions while covering diverse types (forensic detection, emotion recognition, retrieval, visual quality). Using identical hyperparameters with no task-specific tuning, Purified Attention (PurAttn; our purified attention posterior π from Eq. 11, without bias correction) exceeds Vanilla on 5/6 tasks. The strongest is forensic_forgerynet: 86.0% vs 35.0% (+51 points), revealing position bias as a major accuracy bottleneck. RStd decreases across all 6 tasks:
>
> | Task | Vanilla ($\uparrow$) | PurAttn ($\uparrow$) | Ours ($\uparrow$) | $\Delta$ RStd ($\downarrow$)|
> |---|---|---|---|---|
> | forensic_forgerynet | 35.0% | 86.0% | 87.4% | -39.7% |
> | forensic_blink | 24.4% | 27.9% | 30.9% | -11.8% |
> | emotion_expw | 28.4% | 30.8% | 31.8% | -15.4% |
> | emotion_findingemo | 24.3% | 25.8% | 26.9% | -6.0% |
> | text2image_retrieval | 24.3% | 24.6% | 25.2% | -11.6% |
> | visual_quality | 49.8% | 49.0% | 53.0% | -35.0% |
>
> > **W2. Attention faithfulness assumption lacks rigorous validation. (Also addresses Q1)**
>
> We extend Figure 3 to a comprehensive evaluation across the full test set, all GT positions, both N values, and both difficulty settings. Raw Attn denotes the argmax of raw attention weights (Eq. 6); Purified Attn applies prior removal (Eq. 11):
>
> | Setting | Vanilla Acc ($\uparrow$) | Raw Attn ($\uparrow$) | Purified Attn ($\uparrow$) | Ours ($\uparrow$) |
> |---|---|---|---|---|
> | Random N=4 | 67.52% | 91.08% | 96.14% | 98.66% |
> | Random N=8 | 31.56% | 80.78% | 91.16% | 94.92% |
> | Adv N=4 | 49.56% | 42.36% | 64.38% | 71.06% |
> | Adv N=8 | 24.88% | 32.84% | 48.48% | 55.34% |
>
> Purified Attention consistently outperforms Vanilla across all settings, confirming Logit-Attention Divergence is systemic. Vanilla logits show extreme positional dependence (0.00% at position 4 vs 98.90% at position 7 in N=8 random), while position-wise recall under Purified Attention is much more uniform (87–96%). Prior removal (Eq. 11) improves accuracy in every condition (+5.1% to +22.0%).
>
> > **W3. Eq. 2 relies on strong assumptions. (Also addresses Q2)**
>
> We validate Eq. 2 from two complementary angles.
>
> *Residual analysis.* To empirically validate Eq. 2, we measure how strongly position alone explains the variation in non-GT logits using $\eta^2$ (eta-squared), a standard metric of the proportion of variance explained by the position factor. As shown in the table below, before correction, position accounts for a substantial fraction of the variance. After correction, this drops sharply to 2.5% and 8.2%, respectively, showing that positional effects are substantially reduced.
>
>
> | Setting | $\eta^2$ (before) | $\eta^2$ (after) |
> |---|---|---|
> | N=4 | 52.9% | 2.5% |
> | N=8 | 49.6% | 8.2% |
>
> *Independent validation via Perm-Avg.* Perm-Avg cycles GT through all N positions and averages logits, canceling bias by symmetry. Its strong performance (see Q3) empirically supports the additive log-domain structure assumed in Eq. 2.
>
> > **Q1. Systematic attention accuracy analysis.**
>
> Please see W2 above.
>
> > **Q2. Independence assumption validation.**
>
> Please see W3 above.
>
> > **Q3. Why is SoFA excluded? Have you considered permutation averaging?**
>
> We conduct comparisons with SoFA [2] and N-cyclic Perm-Avg on LLaVA-OneVision:
>
> | Setting | Vanilla ($\uparrow$) | SoFA ($\uparrow$) | Perm-Avg ($\uparrow$, N$\times$ cost) | Ours ($\uparrow$) |
> |---|---|---|---|---|
> | Random N=4 | 67.52% | 59.70% | 99.04% | 98.66% |
> | Adv N=4 | 49.56% | 43.26% | 74.32% | 71.06% |
> | Random N=8 | 31.56% | 32.16% | 92.86% | 94.92% |
> | Adv N=8 | 24.88% | 25.34% | 60.16% | 55.34% |
>
> *SoFA.* SoFA underperforms Vanilla in most settings (59.70% vs 67.52% at N=4 random) due to train-test mismatch from modifying attention masks at inference. Our method reads attention without modifying it, and consistently outperforms SoFA across all settings. We will cite and discuss SoFA in later versions.
>
> *Perm-Avg.* Perm-Avg requires N forward passes (4× at N=4, 8× at N=8). Our single-pass method achieves comparable or even superior performance (e.g., 94.92% vs 92.86% at N=8 random) at 1/N the compute cost.
>
> > **Q4. Does Logit-Attention Divergence extend beyond retrieval?**
>
> Please see W1 above.
>
> We thank you for the thorough feedback. We will incorporate these additional results into later versions, and hope they further strengthen the paper and address your remaining comments.
>
> Refs:
> [1] Meng et al. MMIU: Multimodal Multi-image Understanding for Evaluating Large Vision-Language Models. ICLR 2025.
>
> [2] Tian et al. Identifying and Mitigating Position Bias of Multi-Image Vision-Language Models. CVPR 2025.

---

> > ### Author Rebuttal · Reviewer_P5Tm · 2026-04-01
> >
> > The reviewer thanks the authors for their detailed experiments and response, which addressed the reviewer's concerns. Based on the overall quality of the manuscript and the authors' response, the reviewer has decided to maintain the current score.

---

> > > ### Author Response · Authors · 2026-04-01
> > >
> > > Thank you for your thoughtful follow-up and for recognizing that our additional experiments and response have addressed your concerns. We sincerely appreciate your positive assessment of the manuscript and your time.

---

### Official Review · Reviewer_x4Qh · 2026-03-01

**Soundness:** 3
**Presentation:** 3
**Significance:** 2
**Originality:** 3
**Overall Recommendation:** 4
**Confidence:** 3

**Summary:**

This work identifies a phenomenon in MLLMs termed Logit-Attention Divergence, where the model's final output is hijacked by positional priors despite the internal attention mechanisms correctly identifying the target visual evidence. To address this, the authors propose Attention-Guided Debiasing, an inference-time calibration framework that uses internal attention weights to correct biased logits. While the method is training-free and shows improved accuracy on MS-COCO-based benchmarks, its practical utility is constrained by specific architectural requirements.

**Compliance With Llm Reviewing Policy:**

Affirmed.

**Final Justification:**

The authors addressed my concerns about novelty and resolved some of them with additional experiments. I ultimately give a score of 4, but not higher, because the black-box issues they claimed remain unresolved, which I consider important for a comprehensive system. Therefore, my final inclination is borderline accept.

**Key Questions For Authors:**

**Questions**:

1. API Compatibility: Given the white-box requirement, do you see any path toward adapting this "attention-guided" logic for black-box models, perhaps through auxiliary "explainability" outputs?
2. Beyond Retrieval: Position bias is a major issue in long-form generation and multi-image reasoning; can your "Logit-Attention Divergence" insight be applied to stabilize the generation of text rather than just the selection of an index?
3. Encoder Errors: In your adversarial setting, if the CLIP-based hard negatives successfully "confuse" the visual encoder's attention, does the performance of your method collapse?

**Limitations:**

**Limitations**: There remains room for further discussion regarding the research significance and novelty of this work. I invite the authors to provide clarifications on the points raised to better substantiate their contributions.

**Strengths And Weaknesses:**

**Strengths**:
1. Clear Problem Identification: The paper correctly identifies and quantifies the severe position bias inherent in current MLLM autoregressive architectures.
2. Methodological Logic: The transition from observing "conditional position bias" to proposing a Bayesian-inspired mixture prior is technically sound and logically well-motivated.

**Weaknesses**:
1. Practical Applicability Bottleneck: The most significant concern is the requirement for white-box access to internal attention weights. In an era where many state-of-the-art MLLMs are served via black-box APIs, a method that cannot be applied to these models has diminished real-world impact.
2. Narrow Task Scope: The method is strictly tailored for discriminative retrieval where the model chooses from a discrete set of index tokens. It is unclear how this logic would generalize to more complex, open-ended generative tasks that also suffer from position bias.
3. Dependency on Visual Grounding: The framework assumes that internal attention maps are inherently "correct" or "faithful". If the underlying visual encoder fails to ground the query correctly, the calibration mechanism lacks a reliable signal to perform the correction.

---

> ### Author Rebuttal · Authors · 2026-03-29
>
> We thank you for recognizing the clear problem identification and the methodological logic of our work. We address each comment below.
>
> > **W1. White-box access requirement limits practical applicability. (Also addresses Q1)**
>
> We agree that the current framework requires white-box access and is most applicable to open-weight MLLMs. We view this as a limitation of deployment scope rather than of the scientific contribution. Our core finding is that position bias arises primarily as a calibration failure rather than a perception failure, since the model internally attends to the correct image but produces biased outputs. This insight and correction mechanism constitute the main contribution. Adapting to black-box APIs (e.g., via logprob-based bias estimation with proxy attribution signals) is a promising future direction, but is beyond the scope of the current paper.
>
> > **W2. Narrow task scope.**
>
> We provide evidence that Logit-Attention Divergence extends beyond retrieval to discrete-choice multi-image tasks. We select 6 multi-image VQA tasks from MMIU [1] spanning forensic detection (forgerynet, blink), emotion recognition (expw, findingemo), text-to-image retrieval, and visual quality assessment. Their answers are grounded in a finite set of candidate images (e.g., "the first image"), matching the assumptions of our framework. We include Purified Attention (PurAttn; our attention posterior $\pi$ from Eq. 11, used alone without bias correction) to directly test whether Logit-Attention Divergence manifests. Identical hyperparameters, no task-specific tuning:
>
> | Task | Vanilla ($\uparrow$) | PurAttn ($\uparrow$) | Ours ($\uparrow$) | $\Delta$ RStd ($\downarrow$)|
> |---|---|---|---|---|
> | forensic_forgerynet | 35.0% | 86.0% | 87.4% | -39.7% |
> | forensic_blink | 24.4% | 27.9% | 30.9% | -11.8% |
> | emotion_expw | 28.4% | 30.8% | 31.8% | -15.4% |
> | emotion_findingemo | 24.3% | 25.8% | 26.9% | -6.0% |
> | text2image_retrieval | 24.3% | 24.6% | 25.2% | -11.6% |
> | visual_quality | 49.8% | 49.0% | 53.0% | -35.0% |
>
> PurAttn outperforms or matches Vanilla in 5 of 6 tasks, showing that this phenomenon is not limited to retrieval. On forgerynet, this result reveals position bias as a major accuracy bottleneck, suppressing performance from 87.4% to 35.0%. RStd decreases across all 6 tasks, confirming consistent bias mitigation across diverse task types.
>
> > **W3. Method relies on visual grounding, fails if encoder attention is unreliable. (Also addresses Q3)**
>
> We conduct failure analysis on the adversarial set to examine how our method behaves when PurAttn (our purified attention posterior from Eq. 11) fails to identify the correct image. We report two key metrics: how often our bias correction module (Eq. 12--13) still recovers the correct answer despite $\pi$ failure, and how this compares to Vanilla. The last two columns report the percentage of the full evaluation set that is answered correctly despite PurAttn failure.
>
> | Setting | PurAttn failure rate | Ours correct under PurAttn failure	| Vanilla correct under PurAttn failure|
> |---|---|---|---|
> | Adv N=4 | 35.62% | 10.26% | 7.70% |
> | Adv N=8 | 51.52% | 8.84% | 1.60% |
>
> At N=8, where $\pi$ fails on over half the samples, our method still answers 8.84% of all samples correctly, versus 1.60% for Vanilla (+7.24%). This robustness comes from the conditional bias matrix (Eq. 5), which encodes decoder-level structural tendencies independently of visual content and retains corrective power even when the attention signal is uninformative. The cases where both our method and Vanilla fail reflect irreducible perceptual limits where hard negatives defeat the visual encoder entirely. Importantly, such high $\pi$ failure rates (35.62% / 51.52%) are specific to the adversarial setting; under random settings, $\pi$ failure is rare, confirming that attention reliably identifies the correct image under normal conditions.
>
>
> > **Q1. API compatibility path.**
>
> Please see W1 above.
>
> > **Q2. Logit-Attention Divergence beyond retrieval.**
>
> Our VQA experiments (W2) demonstrate that position bias can be the major accuracy bottleneck in discrete-choice multi-image tasks, and our framework effectively addresses this issue. Open-ended generation, where the finite candidate structure is absent, poses additional challenges that we consider a natural extension of this work. However, the core insight that attention remains faithful while outputs are biased could inform generation-time interventions such as attention-guided reranking or constrained decoding.
>
> > **Q3. Encoder-failure analysis.**
>
> Please see W3 above.
>
> We thank you for the constructive feedback. These results, particularly the N=8 analysis showing clear gains under >50% attention failure, further support the effectiveness of our method. We will incorporate them into later versions.
>
> [1] Meng et al. MMIU: Multimodal Multi-image Understanding for Evaluating Large Vision-Language Models. ICLR 2025.

---

> > ### Author Rebuttal · Reviewer_x4Qh · 2026-04-03
> >
> > Thank you to the authors for your efforts in addressing my concerns, and I will therefore raise my score.
> > Good luck!

---

> > > ### Author Response · Authors · 2026-04-03
> > >
> > > Thank you for your thoughtful follow-up and for recognizing that our additional experiments and response have addressed your concerns. We sincerely appreciate your positive feedback.

---

### Official Review · Reviewer_BgyJ · 2026-03-01

**Soundness:** 2
**Presentation:** 2
**Significance:** 3
**Originality:** 2
**Overall Recommendation:** 4
**Confidence:** 3

**Summary:**

This paper addresses the issue of severe position bias in Multimodal Large Language Models (MLLMs) when performing multi-image cross-modal retrieval tasks. Through empirical analysis, the authors identify a phenomenon termed "Logit-Attention Divergence," where the model's internal attention mechanisms correctly locate the relevant visual evidence, but the final autoregressive output logits are corrupted by structural positional priors (e.g., preferring the first or last image). To address this, the authors propose **Attention-Guided Debiasing**, a training-free, inference-time framework. This method utilizes a small calibration set to estimate conditional position bias and leverages the model's intrinsic attention signals to rectify the output distribution.

**Compliance With Llm Reviewing Policy:**

Affirmed.

**Final Justification:**

Refer to my Rebuttal Acknowledgement.

**Key Questions For Authors:**

1.   In Equation 9, you select layers based on attention probability mass. Did you observe significant differences in *which* layers were selected across the different backbones (Qwen2.5-VL vs. LLaVA-OneVision)? Does the "visual evidence" consistently reside in the middle or late layers, and how sensitive is performance to the hyperparameter $K$?
2.  The method uses a small calibration set (e.g., 5 samples). How sensitive is the estimated bias matrix $\hat{P}_{bias}$ to the domain of the calibration images? If the calibration set is visually distinct from the test set (e.g., calibration on diagrams, testing on natural photos), does the "conditional bias" estimation degrade?
3.  While the paper focuses on image retrieval, position bias is also prevalent in multiple-choice Visual Question Answering. Can this framework be directly applied to VQA by treating the options (A, B, C, D) similarly to image indices, or does the text-heavy nature of VQA options introduce different attention dynamics that would invalidate the "Logit-Attention Divergence" hypothesis?

**Limitations:**

yes

**Strengths And Weaknesses:**

**Strengths:**
*    The identification of the "Logit-Attention Divergence" is a significant contribution. The authors explore the aspect of the discrepancy between internal latent focus and final predictive output, providing a convincing explanation that position bias in MLLMs is largely a calibration failure rather than a perception failure.
*   The proposed Attention-Guided Debiasing framework is mathematically well-grounded. By modeling the output probability as a composition of conditional bias and visual evidence, and subsequently using a Bayesian-inspired approach to recover the intrinsic visual probability, the method offers a robust solution.

**Weaknesses:**
*    The method relies on extracting attention weights from specific Transformer layers. This is a significant limitation for closed-source models (e.g., GPT-5, Gemini) where users only have access to API-level logits or text outputs. This restricts the applicability of the method to open-weights models.
*   The current formulation is tailored specifically for discriminative retrieval tasks where the answer space is a closed set of indices. It is not immediately clear how this framework would extend to open-ended generation tasks where position bias might manifest in more subtle ways (e.g., hallucinating details from the last image in a sequence).
*  The adaptive visual evidence selection (Top-K layers) relies on the assumption that layers with high attention concentration are the most informative. While empirically effective, a deeper analysis of *why* certain layers effectively decouple semantics from position bias across different architectures (e.g., Qwen vs. LLaVA) would strengthen the interpretability.

---

> ### Author Rebuttal · Authors · 2026-03-29
>
> We thank you for recognizing the significance of our Logit-Attention Divergence finding and the mathematically well-grounded framework. We address each comment below.
>
> > **W1. Applicability to closed-source models.**
>
> We acknowledge that our current method requires white-box access and therefore targets *open-weight MLLMs*. We view this as a limitation of deployment scope rather than of the core scientific contribution. All three backbones in our paper are open-weight, a standard setting for reproducible MLLM research. Our main contribution is twofold: identifying Logit-Attention Divergence, namely that position bias arises primarily as a *calibration failure* rather than a *perception failure*, and providing a concrete correction mechanism. Extending to black-box APIs is an important future direction, while it lies beyond the scope of the current paper.
>
> > **W2. Extending the framework to open-ended generation tasks. (Also addresses Q3)**
>
> *VQA experiments.* We select 6 multi-image VQA tasks from MMIU [1] spanning forensic detection (forgerynet, blink), emotion recognition (expw, findingemo), text-to-image retrieval, and visual quality assessment. Their answers are grounded in a finite set of candidate images (e.g., "the first image"), matching the assumptions of our current framework. We include Purified Attention (PurAttn; our purified attention posterior from Eq. 11, used alone without bias correction) to directly test whether Logit-Attention Divergence manifests beyond retrieval. Identical hyperparameters, no task-specific tuning:
>
> | Task | Vanilla ($\uparrow$) | PurAttn ($\uparrow$) | Ours ($\uparrow$) | $\Delta$ RStd ($\downarrow$) |
> |---|---|---|---|---|
> | forensic_forgerynet | 35.0% | 86.0% | 87.4% | -39.7% |
> | forensic_blink | 24.4% | 27.9% | 30.9% | -11.8% |
> | emotion_expw | 28.4% | 30.8% | 31.8% | -15.4% |
> | emotion_findingemo | 24.3% | 25.8% | 26.9% | -6.0% |
> | text2image_retrieval | 24.3% | 24.6% | 25.2% | -11.6% |
> | visual_quality | 49.8% | 49.0% | 53.0% | -35.0% |
>
> PurAttn outperforms or matches Vanilla in 5 of 6 tasks, providing broader evidence that Logit-Attention Divergence arises beyond retrieval. On forgerynet, the 51% PurAttn–Vanilla gap is the clearest evidence: attention correctly localizes the target while logits are heavily corrupted, showing that position bias can be a major accuracy bottleneck. RStd decreases across all 6 tasks, confirming consistent bias mitigation.
>
> *Open-ended generation*. Our VQA results above demonstrate that position bias can be the major accuracy bottleneck in discrete-choice multi-image tasks, and our framework effectively addresses this critical issue. Extending to open-ended generation, where the finite candidate structure is absent, is a natural direction for future work.
>
> > **W3. Interpretability of Top-K layer selection across architectures. (Also addresses Q1)**
>
> Performance is robust across $K \in \{1,2,4,8\}$ with $K=2$ optimal (Table 4, §5.6). To examine which layers are selected, we report the top-3 most frequently chosen layers on 1,000 samples across architectures:
>
> | Model | Top-3 selected layers (% of selections) |
> |---|---|
> | LLaVA N=4 | 21 (49.8%), 24 (30.0%), 23 (18.8%) |
> | LLaVA N=8 | 24 (48.3%), 21 (44.3%), 23 (6.9%) |
> | Qwen N=4 | 27 (50.0%), 29 (18.2%), 26 (16.7%) |
> | Qwen N=8 | 27 (49.8%), 29 (31.5%), 33 (12.3%) |
>
> Within each architecture, the same 2–3 layers dominate across N=4 and N=8, confirming stable layer identification. The selected layers differ across architectures, but in both cases cluster in the later half of the network, suggesting useful visual evidence emerges after cross-modal integration.
>
> > **Q1. Cross-architecture layer analysis and K sensitivity.**
>
> Please see W3 above.
>
> > **Q2. Sensitivity to domain gap in calibration images.**
>
> We test extreme domain mismatch: calibrating on ScienceQA diagrams with dummy captions ("A photo.") and evaluating on MS-COCO natural photos. We compare the N×N bias matrices learned from in-domain and cross-domain calibration, and report Spearman ρ, which measures how consistent their positional-bias ordering is across domains:
>
> | Calibration Source | Samples | Acc ($\uparrow$) | Spearman $\rho$ ($\uparrow$) |
> |---|---|---|---|
> | Natural (COCO) | 5 | 98.14% | — |
> | Diagrams (ScienceQA) | 5 | 97.28% | 0.82 |
> | Diagrams (ScienceQA) | 50 | 96.06% | 0.89 |
>
> Accuracy drops by only 0.86% under extreme visual domain mismatch, while the positional-bias ordering remains highly consistent across domains. This supports our claim (§4.2) that the bias matrix captures decoder-level structural tendencies that are largely invariant to visual content.
>
> > **Q3. Applicability to VQA**
>
> Please see W2 above.
>
> We thank you for the constructive feedback. We hope these additions address your comments. We will incorporate them into later versions.
>
> [1] Meng et al. MMIU: Multimodal Multi-image Understanding for Evaluating Large Vision-Language Models. ICLR 2025.

---

> > ### Author Rebuttal · Reviewer_BgyJ · 2026-04-01
> >
> > I think authors solved my concerns, so I am glad to raise my score from 3 to 4.

---

> > > ### Author Response · Authors · 2026-04-01
> > >
> > > Thank you for your thoughtful follow-up and for recognizing that our rebuttal has addressed your concerns. We sincerely appreciate your positive feedback.

---

### Official Review · Reviewer_tAdC · 2026-03-13

**Soundness:** 4
**Presentation:** 4
**Significance:** 3
**Originality:** 3
**Overall Recommendation:** 4
**Confidence:** 3

**Summary:**

This paper investigates the position bias of multimodal large language models (MLLMs) in multi-image retrieval tasks. In this task, the model receives a text query and multiple candidate images in order, and needs to output the index position of the correct image. The authors find that current MLLMs often suffer from significant position bias: the model's predictions are significantly influenced by the order in which candidate images are arranged, rather than being based solely on semantic matches. The paper proposes and analyzes an interesting phenomenon called Logit-Attention Divergence. In particular, the final logit prediction of the model tends to be biased towards certain fixed locations, even if the internal attention distribution of the model is correctly focused on the target image. This indicates that the model may have completed visual semantic matching internally, but the final decision layer is still affected by position bias. Based on this observation, the authors propose a calibration method for the inference phase of Attention-Guided Debiasing. This is a training-free inference time calibration method, which effectively alleviates the position bias in multi-image retrieval.

**Compliance With Llm Reviewing Policy:**

Affirmed.

**Final Justification:**

I maintain my original rating.

**Key Questions For Authors:**

1. How sensitive is the method to different prompt templates?
2. Under what circumstances does this approach break down, such as when attention itself becomes unreliable?

**Limitations:**

Please see weeknesses.

**Strengths And Weaknesses:**

Pros:
1. This work focuses on an important but less studied problem in MLLMs. position bias is a practical and significant issue in multi-graph input scenarios.
2. This work presents a clear and illuminating phenomenon: Logit-Attention Divergence.
3. The proposed attention-guided calibration method is training-free, and only a minimal amount of calibration data is required.
4. The design of Conditional bias modeling is reasonable.

Cons:
1. The probabilistic decomposition of Eq. 2 in the paper is mainly based on intuitive assumptions, rather than strictly derived from the model structure or training objective.
2. The experiments focus on multi-image retrieval/image selection task. It is not clear whether the method can be generalized to other tasks.
3. It would be better to have code attached to this paper, as the absence of code at this time would affect the judgment of its reproducibility.

---

> ### Author Rebuttal · Authors · 2026-03-29
>
> We thank you for recognizing the important problem, the clear Logit-Attention Divergence phenomenon, the training-free method with minimal calibration data, and the reasonable conditional bias modeling design.
>
> > **W1. The probabilistic decomposition of Eq. 2 is based on intuitive assumptions.**
>
> Our factorization adopts the same log-domain additive assumption as PriDe [1], widely accepted in the calibration literature. Deriving position bias correction from first principles remains an open problem in the field.
>
> To empirically validate Eq. 2, we measure how strongly position alone explains the variation in non-GT logits using $\eta^2$ (eta-squared), a standard metric of the proportion of variance explained by the position factor. As shown in the table below, before correction, position accounts for a substantial fraction of the variance. After correction, this drops sharply to 2.5% and 8.2%, respectively, showing that positional effects are substantially reduced.
>
> | | $\eta^2$ (before) | $\eta^2$ (after) |
> |---|---|---|
> | N=4 | 52.9% | 2.5% |
> | N=8 | 49.6% | 8.2% |
>
>
> > **W2. Generalization beyond multi-image retrieval.**
>
> To test generalization, we select 6 multi-image VQA tasks from MMIU [2] spanning forensic detection (forgerynet, blink), emotion recognition (expw, findingemo), text-to-image retrieval, and visual quality assessment. Their answers are grounded in candidate images (e.g., "the first image"), matching the assumptions of our framework. Using identical hyperparameters (no task-specific tuning):
>
> | Task | Vanilla→Ours Acc ($\uparrow$) | $\Delta$ RStd ($\downarrow$) |
> |---|---|---|
> | forensic_forgerynet | 35.0→87.4% | -39.7% |
> | forensic_blink | 24.4→30.9% | -11.8% |
> | emotion_expw | 28.4→31.8% | -15.4% |
> | emotion_findingemo | 24.3→26.9% | -6.0% |
> | text2image_retrieval | 24.3→25.2% | -11.6% |
> | visual_quality | 49.8→53.0% | -35.0% |
>
> Across all 6 MMIU tasks, our method consistently reduces RStd, indicating that it mitigates position bias beyond pure retrieval settings. Accuracy also improves on every task without any task-specific tuning, suggesting our method extends to a broader family of discrete-choice multi-image understanding tasks.
>
> > **W3. Code availability.**
>
> We will release code for reproducing all experiments in the paper.
>
> > **Q1. Sensitivity to prompt templates.**
>
> We evaluate three prompt variants on LLaVA-OneVision (N=4): "Default" (our standard template, Table 6), "Minimal" (simplified version of Default), and "Rephrased" (same information as Default with varied wording and sentence order).
>
> | Prompt | Vanilla Acc ($\uparrow$) | Ours Acc ($\uparrow$) | RStd ($\downarrow$) |
> |---|---|---|---|
> | Default | 67.52% | 98.66% | 0.88 |
> | Minimal | 65.44% | 96.86% | 2.48 |
> | Rephrased | 69.26% | 97.94% | 0.88 |
>
>
> For each prompt variant, we evaluate 5 random input permutations (as in §5.2), yielding $3 \times 5 = 15$ matched accuracy pairs. Paired t-tests across all prompts confirm that our method significantly outperforms Vanilla in Random setting (p<0.001).
>
> > **Q2. When does the approach break down?**
>
> We conduct systematic failure analysis on the adversarial set to examine when our method breaks down, namely when PurAttn  $\pi$ (our attention module, Eq. 11) fails to identify the correct image. We report two key metrics: how often our bias correction module (Eq. 12--13) still recovers the correct answer despite $\pi$ failure, and how this compares to Vanilla. The last two columns report the percentage of the full evaluation set that is answered correctly despite PurAttn failure.
>
> | Setting | PurAttn failure rate | Ours correct under PurAttn failure | Vanilla correct under PurAttn failure |
> |---|---|---|---|
> | Adv N=4 | 35.62% | 10.26% | 7.70% |
> | Adv N=8 | 51.52% | 8.84% | 1.60% |
>
> At N=8, PurAttn fails on 51.52% of samples. Even then, our full method still answers 8.84% of the full evaluation set correctly, compared to 1.60% for Vanilla (+7.24%). This robustness comes from the conditional bias matrix (Eq. 5), which retains corrective power even when attention is uninformative. The cases where both our method and Vanilla fail reflect irreducible perceptual limits where hard negatives defeat the visual encoder entirely. Importantly, such high $\pi$ failure rates (35–52%) are specific to the adversarial setting; under random settings, $\pi$ failure is rare, confirming that attention reliably identifies the correct image under normal conditions.
>
> We thank you for the valuable feedback. These results, including VQA generalization, t-test validation, and failure analysis, further support the effectiveness of our method. We will incorporate them into later versions.
>
> [1] Zheng et al. Large Language Models Are Not Robust Multiple Choice Selectors. ICLR 2024.
>
> [2] Meng et al. MMIU: Multimodal Multi-image Understanding for Evaluating Large Vision-Language Models. ICLR 2025.

---

> > ### Author Rebuttal · Reviewer_tAdC · 2026-04-03
> >
> > Thank you for the detailed response! I recommend including these details in future versions. I will keep my initial positive assessment unchanged.

---

> > > ### Author Response · Authors · 2026-04-03
> > >
> > > Thank you for your thoughtful follow-up and for recognizing that our rebuttal has addressed your concerns. We truly appreciate your helpful suggestion. We will incorporate these additional details into future versions of the paper.

---

### Decision · Program_Chairs · 2026-04-30

**Decision:**

Accept (regular)

**Comment:**

The paper proposes a training-free calibration approach to mitigate the position bias issue in multi-image retrieval.

Some core strength of the submission include:
- tackles an unstudied problem of position bias in MLLMs for multi-image retrieval
- the attention guided calibration method is training free
- attention-guided framework is mathematically grounded
- shows good empirical results and comprehensive  evaluation design

Among the major weaknesses noted by the reviewers include:
- generalizability of the method to other tasks
- unclear applicability to closed-source models
- reliance of method on visual grounding
- factorization model is based on strong assumptions

In the post-rebuttal phase, all reviewers remained positive about the submission, indicating that their major concerns have been addressed satisfactorily. The final rating from all reviewers was weak accept. As mentioned by one reviewer that the applicability to only open-weight models could be a limiting factor. However, as noted by reviewers that the method is novel, with proper motivation and strong results so this applicability point should not be a point of rejection. Also, all other concerns have been well-resolved and acknowledged by all the reviewers. As it stands, the AC decides to recommend acceptance and encourage authors to include important points from rebuttal and post-discussion into the final version.